# Acoustofluidic Interfaces for the Mechanobiological Secretome of MSCs

Ye He[1], Shujie Yang [1], Pengzhan Liu [1], Ke Li [1], Ke Jin[1], Ryan Becker [2], Jinxin Zhang[1], Chuanchuan Lin[3], Jianping Xia [1], Zhehan Ma[2], Zhiteng Ma[1], Ruoyu Zhong[1], Luke P. Lee [4,5,6,7,8] ✉ & Tony Jun Huang [1] ✉

While mesenchymal stem cells (MSCs) have gained enormous attention due to their unique properties of self-renewal, colony formation, and differentiation potential, the MSC secretome has become attractive due to its roles in immunomodulation, anti-inflammatory activity, angiogenesis, and anti-apoptosis. However, the precise stimulation and efficient production of the MSC secretome for therapeutic applications are challenging problems to solve. Here, we report on Acoustofluidic Interfaces for the Mechanobiological Secretome of MSCs: AIMS. We create an acoustofluidic mechanobiological environment to form reproducible three-dimensional MSC aggregates, which produce the MSC secretome with high efficiency. We confirm the increased MSC secretome is due to improved cell-cell interactions using AIMS: the key mediator N-cadherin was up-regulated while functional blocking of N-cadherin resulted in no enhancement of the secretome. After being primed by IFN-γ, the secretome profile of the MSC aggregates contains more anti-inflammatory cytokines and can be used to inhibit the pro-inflammatory response of M1 phenotype macrophages, suppress T cell activation, and support B cell functions. As such, the MSC secretome can be modified for personalized secretome-based therapies. AIMS acts as a powerful tool for improving the MSC secretome and precisely tuning the secretory profile to develop new treatments in translational medicine.

Mesenchymal stem cells (MSCs) are common multipotent stem cells present in many tissues and organs, such as bone marrow, adipose tissue, and umbilical cords[1–3]. Although the multilineage differentiation potential of MSCs has been long regarded as their primary contribution to regenerative medicine, current research has shown that the therapeutic potential of MSCs is attributed mainly to secretome-mediated paracrine effects[4,5]. Rather than solely acting as the building blocks during tissue regeneration, MSCs are now also regarded as a "drug reservoir" that secretes various biomolecules to regulate physiological events[6–9]. The MSC secretome refers to a broad spectrum of secreted trophic molecules such as growth factors, chemokines, and inflammatory factors released from MSCs as soluble proteins or packaged exosomes to participate in immune responses, wound repair, and tissue regeneration processes[10–12]. Compared with

[1]Thomas Lord Department of Mechanical Engineering and Materials Science, Duke University, Durham, NC 27708, USA. [2]Department of Biomedical Engineering, Duke University, Durham, NC 27708, USA. [3]Department of Blood Transfusion, Irradiation Biology Laboratory, Xinqiao Hospital, Chongqing 400037, China. [4]Harvard Medical School, Harvard University, Renal Division and Division of Engineering in Medicine, Department of Medicine, Brigham and Women's Hospital, Boston, MA 02115, USA. [5]Department of Bioengineering, University of California, Berkeley, Berkeley, CA 94720, USA. [6]Department of Electrical Engineering and Computer Science, University of California, Berkeley, Berkeley, CA 94720, USA. [7]Department of Biophysics, Institute of Quantum Biophysics, Sungkyunkwan University, Suwon, Korea. [8]Department of Chemistry and Nanoscience, Ewha Womans University, Seoul, Korea. ✉e-mail: lplee@bwh.harvard.edu; tony.huang@duke.edu

traditional stem cell therapy, the secretome is acellular, which affords several unique advantages for regenerative medicine, such as eliminating ethical issues and tumorigenicity regarding the clinical use of stem cells[13,14]. Additionally, collecting secretome products is much easier than collecting and culturing endogenous stem cells for patient-specific therapies and can be done in significantly larger quantities[15]. The secretome profile can also be flexibly customized according to the specific need or treatment. Therefore, developing a practical method for rapidly and efficiently improving the MSC secretome holds significant value for clinical therapies.

MSCs have shown mechanosensitivity in tissue remodeling, wound repair, and organ development, where external mechanical cues can trigger significant cell reactions through several mechanosensing pathways[16–19]. Cell-extracellular matrix (ECM) and cell-cell interactions are well-known mediators of mechanical forces experienced by MSCs, regulating their secretory properties[20–23]. Currently, methods to improve the MSC secretome mainly take advantage of cell-ECM/cell-cell interactions by developing new biomaterial and microfluidic-based technologies[20–23]. For example, hydrogels have been modified with functional molecules (such as cell adhesion peptides) to mediate cell-matrix interactions, thereby influencing downstream signaling and cell functions[21]. Hydrogels/scaffolds can also be fabricated with tunable physical parameters (Young's modulus, viscoelasticity, porosity, etc.) to influence cell growth and morphology and modify their protein secretion profiles[24–26]. MSCs can also be 3D-cultured in microfluidic devices to improve cell-cell interactions and enhance their protein secretion[27–29]. However, these technologies are very complex, involving many complicated fabrication steps. Additionally, MSCs are highly sensitive to their environment, and when cultured under specific conditions, such as within microfluidic channels and hydrogel droplets, it may be difficult to replicate their typical growth environment. As a result of these methods, chemical contaminants are present in the growth environment and can affect the physiological status and functions of the MSCs. Thus, developing secretome production strategies that preserve the natural growth conditions of MSCs is critical to generating high-quality secretome profiles that are sufficient for therapeutic use.

Acoustofluidic technologies have proven to be a powerful method to manipulate biological objects because of their high biocompatibility, simplicity, and versatility[30–40]. Acoustic waves can be directly transmitted into a liquid medium to generate an acoustic radiation force or acoustic streaming force that can be used to manipulate extracellular vesicles, cells, spheroids, and even small organisms in a non-contact and label-free manner[41–49]. Compared with the technologies mentioned above, acoustofluidics offer a simple, biocompatible, and effective ECM-free method to manipulate MSCs under natural growth conditions. Here, we designed acoustofluidic interfaces for the mechanobiological secretome of MSCs (AIMS). Within the AIMS platform, 3D MSC aggregates can be formed rapidly and yield high-quality cell–cell interactions due to the layered patterning and tight assembly of the cells. Additionally, protein secretion and exosome production of MSCs after acoustofluidic assembly and mechanobiological stimulation are significantly increased. We also found that the expression of N-cadherin was up-regulated, demonstrating that the improved MSC secretome was due to enhanced cell-cell interactions. To demonstrate the therapeutic potential of our AIMS platform, we used the acoustic mechanobiological treatment to tune the immunomodulation activity of MSCs, producing secretome profiles with potent immunomodulatory properties.

## Results

### The acoustofluidic mechanobiological assembly formed MSC aggregates

With the AIMS platform, we used acoustic streaming to assemble MSCs spheroids. We hypothesized that forming 3D MSC aggregates would enhance cell-cell interactions in an ECM-free manner, thus improving the MSC secretome (Fig. 1a, c). A piezoelectric ring was attached to a thin glass coverslip and bonded to a single-well plate using epoxy (Permatex, USA). The piezoelectric ring generates circular standing flexural waves when a sinusoidal voltage excitation signal is applied, producing a gentle and steady acoustic streaming effect in the fluid. According to simulations of the device, as shown in Fig. 1b, a high-intensity focal point is formed in the center of the substrate. The acoustic waves leak into the droplet, creating a

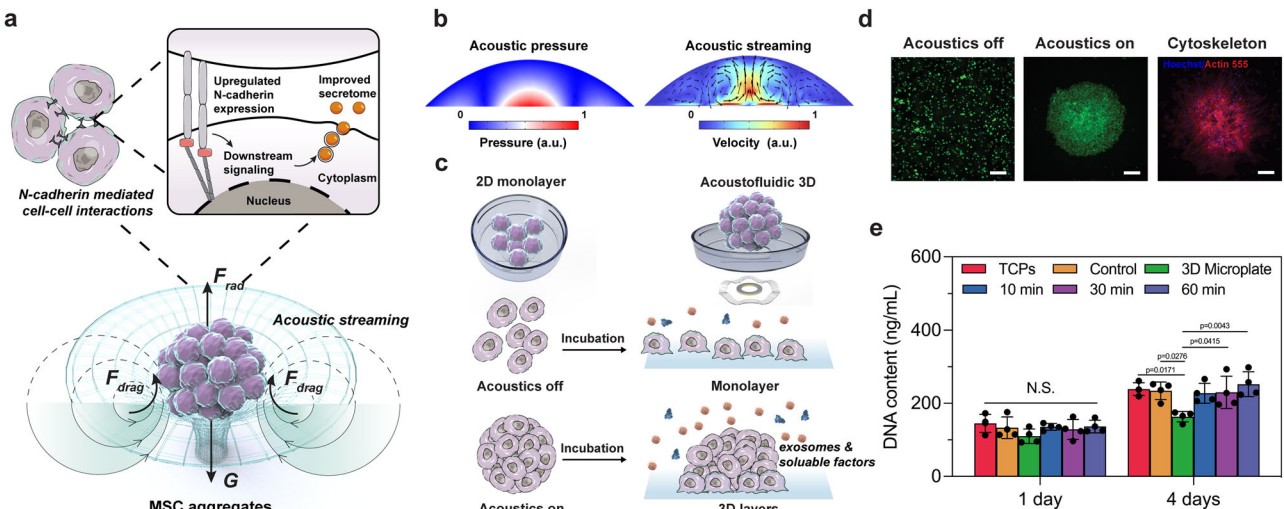

**Fig. 1 | Schematic of the improved stem cell secretome via acoustofluidic mechanobiological assembly. a** Schematic of the formation of MSC aggregates via acoustofluidic assembly and N-cadherin mediated cell–cell interactions to enhance the MSC secretome. **b** The simulation results showed that a three-dimensional circular vortex tube generated by a high-intensity focal point at the center of the substrate functioned as a virtual wall, trapping and aggregating cells at the center of the droplet. **c** Illustration of the comparison between 2D monolayer MSC culture and acoustofluidic 3D MSC aggregation. **d** Fluorescence images of MSCs before/after the acoustofluidic assembly, and the morphology of MSC aggregates after 3 days of incubation. The green color indicates alive cells. The blue color indicates cell nuclei, and the red color indicates the cytoskeleton. $n = 4$ tests with similar results. Scale bar: 100 μm. **e** CyQUANT™ Cell Proliferation Assay measuring the DNA contents of different groups. The statistical analysis was performed using one-way ANOVA with Tukey's post-hoc test. Data are graphed as the mean ± SD ($n = 4$, biological repeats). Source data are provided as a Source Data file.

circular vortex via acoustic streaming. The three-dimensional circular vortex functions as a virtual wall trapping and concentrating the cells into the center of the droplet. After adding the MSCs into the droplets, cells are rapidly driven into the central region to form cell aggregates within 90 s. The MSC aggregates form a spherical shape (Fig. 1d). Because our AIMS platform is a non-contact, ECM-free method for conditioning the MSCs, we anticipated that the formed 3D MSC aggregates would yield improved secretome profiles. To better distinguish the potential effects of 3D cell structure and acoustofluidic stimulation, we also deployed commercialized Spheroid Microplates to 3D-culture MSCs and detected their secretion profile.

To investigate whether different time durations of acoustofluidic assembly would affect the quality of the secretome profile, AIMS treatments were performed for 10, 30, and 60 min, respectively. To prevent premature cell adhesion, the device substrate was pre-treated with polyvinyl alcohol (Mowiol® 18-88). This allowed cells to remain suspended for 24 h, further improving the formation and duration of the cell aggregates. After 24 h of incubation, the aggregates gradually adhered to the substrates. After 3 days of incubation, the MSCs spread onto the substrate, and the pseudopodium and cytoskeleton were observed (Fig. 1d). The cell aggregates showed a clear 3D morphology and tight cell-cell junctions. After 3 days, the aggregate morphology did not show clear changes. To determine the biocompatibility of our AIMS method, the cell proliferation after acoustofluidic assembly was investigated. Cells cultured on cell culture dishes without polyvinyl alcohol treatment or acoustofluidic assembly were used as a positive control (denoted as TCPs in Fig. 1e). Cells cultured in the commercialized Spheroid Microplates were denoted as the 3D Microplate group. We used CyQUANT™ Cell Proliferation Assay to directly measure the total DNA content of each group. As shown in Fig. 1e, after 1 day of incubation, there was no significant difference in DNA content among all groups. However, after 4 days of incubation, the DNA content of the 3D Microplate group was lower than other groups, which meant that

the commercialized Spheroid Microplates were not suitable for long-term MSC proliferation.

## MSC secretome was improved after acoustofluidic assembly

To verify whether the formed cell aggregates increased the production of secretome products after acoustofluidic assembly, cytokine secretion and exosome production levels were investigated (Fig. 2a). To eliminate the latent error brought from the cell number difference of different groups (Fig. 1e), the results of each group in the following experiments were normalized to the constant DNA content (1 µg). The concentrations of a broad spectrum of soluble proteins comprised of 80 total targets (78 available), such as growth factors and chemokines, were investigated using a cytokine assay to analyze the cytokine secretion profiles (Fig. 2b). For the 60 min AIMS treatment group, 57 targets had the highest concentration compared to other groups, and 14 targets had the second-highest concentration. For the 30 min AIMS treatment group, 14 targets had the highest concentration, and 44 targets had the second-highest concentration. However, for the 10 min AIMS treatment group, only 6 targets had the highest concentration and 10 targets had the second-highest concentration. As for 3D Microplate group, after normalizing to the same DNA content, only 1 target had the highest concentration and 9 targets had the second-highest concentration. In addition, 65 targets had the lowest concentration in the control group. In general, the 3D Microplate group and the 10 min, 30 min, and 60 min AIMS treatment groups improved the secretome profile of MSCs as compared to the control group. The 60 min AIMS treatment group had the best effect. However, it is also worth noting that the original integrated density of these four groups only had slight differences (Supplementary Fig. 1), which indicated that their secretome profiles were relatively similar.

To evaluate the secretome improvement effect quantitatively and more accurately, we selected eight of the most representative growth factors and cytokines, FGF-2, HGF-1, VEGF, HGF-1, IL-6, IL-10, IFN-γ, and TNF-α, and detected their secretion levels using an ELISA assay. All the

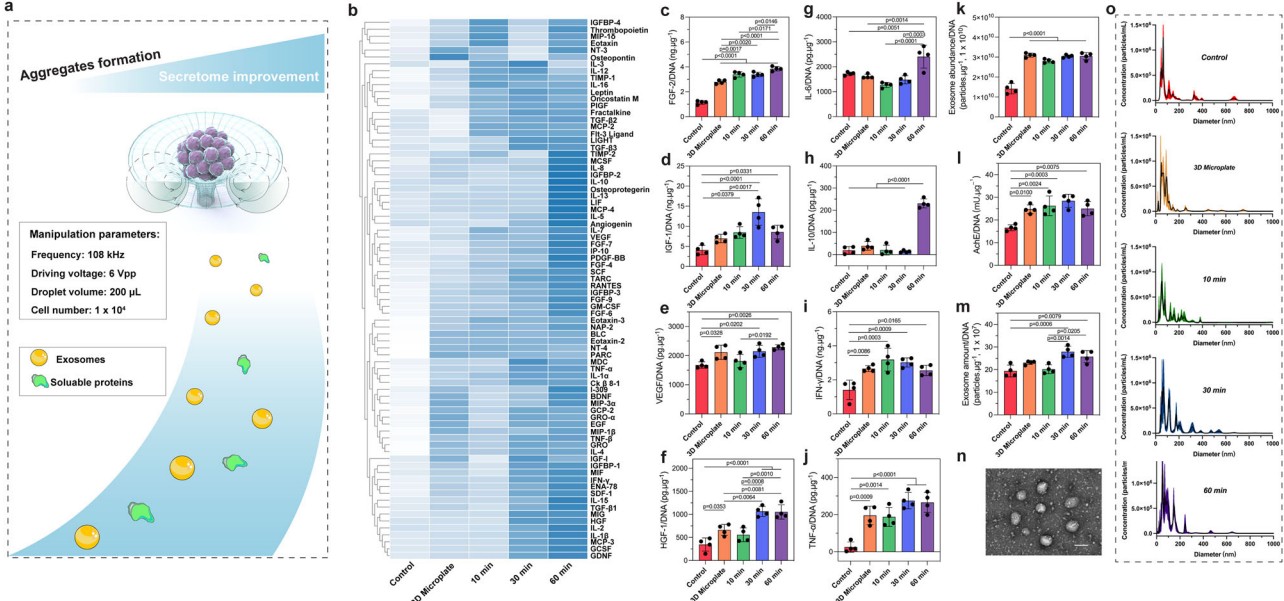

**Fig. 2 | MSC secretome enhancement via AIMS. a** Schematic of the secretome enhancement via AIMS. **b** Heatmap presenting the secretome profiles of different groups after incubating for 4 days. The data was processed by the Z-score normalization and clustering analysis. **c–j** ELISA assay measuring the secretion of FGF-2, IGF-1, VEGF, HGF-1, IL-6, IL-10, IFN-γ, and TNF-α from different groups after incubating for 4 days. **k** The amounts of secreted exosomes from different groups measured by the EXOCET Exosome Quantitation Kit. **l** AchE activity of different groups after incubating for 4 days. **m** Secreted exosome numbers of different groups analyzed using NTA. **n** A representative TEM image of secreted exosomes from MSC aggregates. *n* = 3 tests. Scale bar: 100 nm. **o** NTA test measuring the diameters of secreted exosomes from different groups. The statistical analysis was performed using one-way ANOVA with Tukey's post-hoc test. Data are graphed as the mean ± SD (*n* = 4, biological repeats). Source data are provided as a Source Data file.

results were normalized to DNA content. As shown in Fig. 2c–j, for FGF-2, the 3D Microplate group and the 10 min, 30 min, and 60 min AIMS treatment groups showed increases in concentration compared to the control group. And the concentrations of the 10 min and 30 min AIMS treatment groups were higher than the 3D Microplate group, but lower than that of the 60 min group. For IGF-1, the concentrations between the control and 3D Microplate groups had no significant difference, but the groups of 10 min, 30 min, and 60 min AIMS treatment were higher than that of the control group. For VEGF, the 3D Microplate group and the 30 min and 60 min AIMS treatment groups showed increases in concentration compared to the control group. Similarly, the 3D Microplate group and the 30 min and 60 min AIMS treatment groups showed increases in HGF-1 concentration compared to the control group. The 30 min and 60 min AIMS treatment groups also showed higher concentrations compared to the 3D Microplate and 10 min AIMS treatment groups. For IL-6, the 60 min AIMS treatment group had the highest concentration compared to the control, 3D Microplate, 10 min, and 30 min AIMS treatment groups. As for IL-10, similarly, the 60 min AIMS treatment group had the highest concentration compared to other groups. For IFN-γ, the 3D Microplate and the 10 min, 30 min, and 60 min AIMS treatment groups showed increases in concentration compared to the control group. At last, for TNF-α, the 3D Microplate and the 10 min, 30 min, and 60 min AIMS treatment groups were still higher than that of the control group. Taken together, the ELISA assay results were largely consistent with the cytokine array assay. All the targets measured (including growth factors, chemokines, or inflammatory factors) had higher concentrations in the 3D Microplate and the AIMS treatment groups. For some of the targets (such as FGF-2, HGF-1, IL-6, and IL-10), the 60 min group had the highest concentrations.

Next, the exosome production of each group was investigated. As shown in Fig. 2k, all of the AIMS treatment groups (10, 30, and 60 min)

and the 3D Microplate group exhibited statistically higher exosome levels than the control group. These results were further verified by directly measuring the acetylcholinesterase (AchE) activity, as AchE activity is strongly related to exosome abundance. As shown in Fig. 2l, AchE activity in the AIMS treatment groups and the 3D Microplate group was still significantly higher than in the control group. Subsequently, nanoparticle tracking analysis (NTA) was used to study the exosome abundance (Fig. 2m). Here we found the control group, the 3D Microplate group, and the 10 min AIMS treatment group had no significant difference, but they were lower than the 30 min and 60 min AIMS treatment groups. The exosome characterizations, including morphology and diameter measurement, are shown in Fig. 2n, o. The exosomes collected following acoustofluidic assembly showed a typical globular shape, and their diameters were within the 50–150 nm range. According to our results, both the 3D Microplate and the AIMS treatment groups upregulated the overall MSC secretome. The 3D Microplate group showed comparable effects with the 10 min AIMS treatment group. Therefore, with a longer time duration, especially 60 min, the AIMS treatment can further enhance the MSC secretome.

## The Improved MSC secretome was attributed to increased cell–cell interactions mediated by N-cadherin

To validate the hypothesis that improved cell-cell interactions due to the formation of ECM-free 3D cell aggregates were responsible for the improved MSC secretome using the AIMS platform, N-cadherin activity of the AIMS confined cell aggregates was investigated (Fig. 3a). N-cadherin is well established as a critical mediator of cell-cell interactions. More importantly, several studies have claimed that N-cadherin can amplify the paracrine effects of stem cells[50,51]. Motivated by these results, we investigated the expression and functionality of N-cadherin after the AIMS assembly of the MSC spheroids. We used an N-cadherin neutralizing antibody (nAb) to observe if

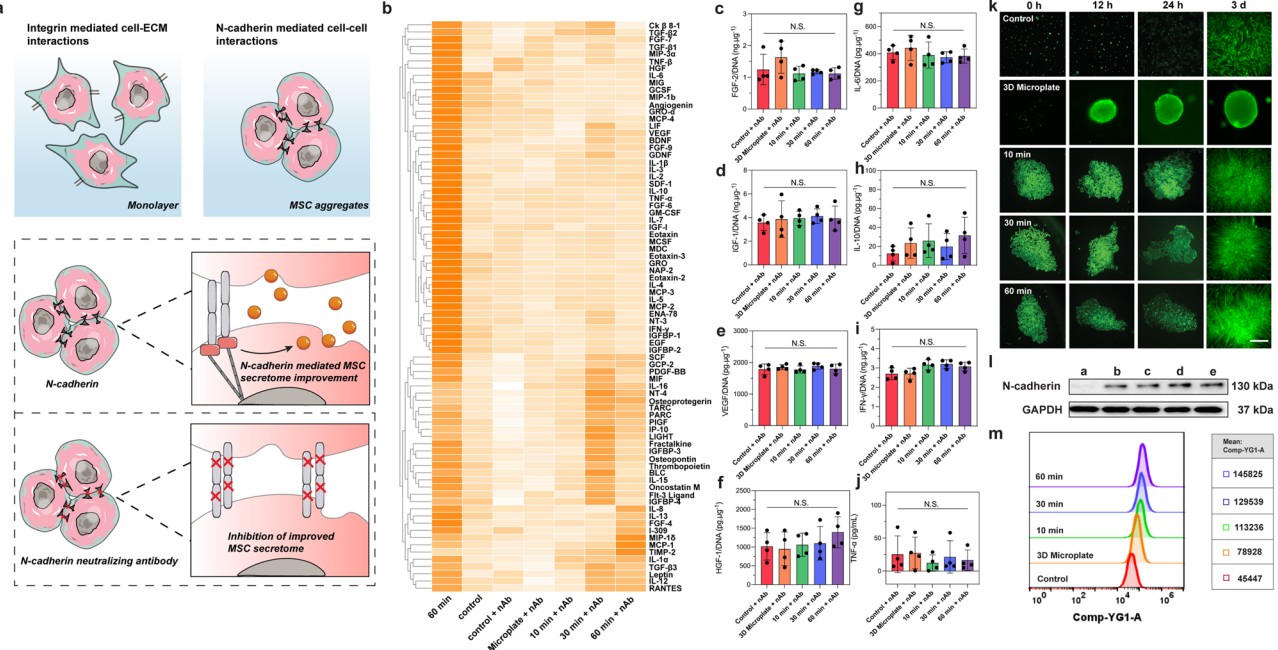

**Fig. 3 | The enhanced MSC secretome was attributed to increased cell-cell interactions mediated by N-cadherin. a** Schematic illustrating how N-cadherin mediated cell-cell interactions contribute to the enhancement of the MSC secretome. **b** Heatmap measuring the secretome profiles of different groups after functional blocking of N-cadherin of MSCs and incubating for 4 days. The data was processed by the Z-score normalization and clustering analysis. **c–j** ELISA assay measuring the secretion of FGF-2, IGF-1, VEGF, HGF-1, IL-6, IL-10, IFN-γ, and TNF-α from different groups after functional blocking of N-cadherin of MSCs and incubating for 4 days (n = 4 tests, biological repeats). **k** Representative FDA staining images of different groups after incubating for 0, 12, 24 h, and 3 days, respectively. n = 3 tests with similar results. Scale bar: 100 μm. **l** Western blotting detecting the expression of N-cadherin of MSCs from different groups. n = 3 tests. a: control, b: 3D Microplate, c: 10 min, d: 30 min, e: 60 min. **m** Flow cytometry analysis measuring the expression of N-cadherin of MSCs from different groups. The statistical analysis was performed using one-way ANOVA with Tukey's post-hoc test. Data are graphed as the mean ± SD. Source data are provided as a Source Data file.

functionally blocking N-cadherin affected the secretome profile. After pre-treating with nAb, we performed 3D Microplate culture and AIMS assembly for the cells. We also used normal MSCs without nAb treatment as the control and 60 min AIMS treatment groups. Similar to above, the total DNA content of each group was measured and used for data normalization (Supplementary Fig. 2). Firstly, we conducted the cytokine array assay again (79 targets available), with the results shown in Fig. 3b. Similar to previous results, the normal control group and the 60 min AIMS treatment group showed a clear difference. However, for the nAb pre-treated groups, it is evident that the cytokine secretion profiles of each group show no significant changes, unlike the results obtained in Fig. 2b that contained active N-cadherin, suggesting that the AIMS-induced enhancements to the MSC secretome were eliminated after functionally blocking N-cadherin. The original data (heatmap without z-score normalization) showed that the integrated densities of each group were similar (Supplementary Fig. 3), although some of the targets in 30 min AIMS treatment + nAb and 60 min AIMS treatment + nAb were slightly higher than other nAb pre-treated groups. Next, we repeated the ELISA assay for the 8 targets to examine whether the improved secretome was abolished. As shown in Fig. 3c–j, same to cytokine array assay, the cytokine secretion had the similar tendency among all groups.

Next, we investigated the expression differences of N-cadherin among all groups. At first, we observed the cell-cell interactions in different groups because N-cadherin is the main mediator. As shown in Fig. 3k, for the 3D Microplate group, because there was a hydrophilic hydrogel coating on the substrate, the cells maintained a suspended status for all time. In other groups, due to the polyvinyl alcohol pretreatment, MSCs also maintained a suspended status after 24 h of incubation. However, the shapes of the MSC aggregates in the 10, 30, and 60 min AIMS treatment groups showed little difference. In most cases, the morphology of the 3D cell aggregates could be well maintained when they were suspended for the first 24 h. However, some of the cell aggregates from the 10 and 30 min AIMS treatment groups were dispersed after 24 h, which may be attributed to insufficient acoustofluidic assembly durations. Only the 60 min AIMS treatment group showed a stable morphology of the 3D cell aggregates. These results suggest that different time durations of AIMS treatment may influence the morphology and maintenance of MSC 3D cell aggregates. After 3 days, MSCs in all groups were adhered to the substrates, as the pseudopodium and cytoskeleton were observed. Coupled cell layers were stacked after AIMS treatment. However, the control group presented a normal monolayer morphology. These results made clear that after the AIMS treatment, MSCs showed increased cell–cell interactions. Then we used Cyto-immunofluorescence staining to visualize the N-cadherin expression. As shown in Supplementary Fig. 4, after 3 days of incubation, we noticed that the fluorescence intensity of N-cadherin in the 10, 30, and 60 min AIMS treatment groups was higher than that of the control group. We also used Western blotting assay and flow cytometry assay to measure the N-cadherin expression. As shown in Fig. 3l, m, in general, the 3D Microplate group and the AIMS treatment groups (10 min, 30 min, and 60 min) had a clear increase of N-cadherin expression than the control group. Flow cytometry showed that the 60 min AIMS treatment group had the highest N-cadherin expression level. The 3D Microplate group and the 10 min and 30 min AIMS treatment groups also had higher levels than the control group. We also use QPCR assay to detect N-cadherin expression at the genetic level. As shown in Supplementary Fig. 5, although all the 3D Microplate and the AIMS treatment groups still had more CDH2 (N-cadherin) gene expressions than the control group, these four groups had no significant difference. Even though there was some inconsistency between the genetic and protein level, it is still reasonable to conclude that the formation of the MSC spheroids up-regulates the expression of N-cadherin, improving the secretome profile of the MSCs.

Furthermore, it is well known that external mechanical stresses may cause cells to initiate their protection and repair mechanisms. This will trigger a $Ca^{2+}$ influx and stimulate the endosomal release of cells. A previous study also reported that after high-frequency acoustic stimulation[52], the exosome production of U87-MG cells was increased due to the increased $Ca^{2+}$ influx. Therefore, we investigated the intracellular $Ca^{2+}$ concentrations and ATP levels to observe whether the acoustofluidic mechanobiological stimulus was a source of significant mechanical stress to the MSCs. As shown in Supplementary Figs. 6 and 7, both the intracellular $Ca^{2+}$ concentration and ATP level had no statistical difference among all groups, indicating that the AIMS treatment was gentle and that the enhanced secretome profile in this study was neither related to the changes in intracellular $Ca^{2+}$ concentration nor ATP level.

## Harnessing the AIMS-enhanced MSC secretome for immunomodulation applications

MSC secretome have long been shown to have immunomodulatory functions[11,53] due to the large collection of anti-inflammatory and growth factors they secrete[54]. However, their efficiency is significantly limited by the cell dose and activity[11]. Because AIMS was able to up-regulate the overall MSC secretome, we explored the possibility of using AIMS to tune the immunomodulation function of MSCs. According to previous studies, MSCs were primed by IFN-γ to help convert them into an immunomodulatory phenotype[55]. AIMS was then performed on the IFN-γ primed MSCs to form MSC aggregates. The 60 min AIMS treatment group was selected in this section because they have a more stable cell aggregate shape and an improved MSC secretome. Firstly, the DNA content of each group was studied and used for data normalization (Supplementary Fig. 8). Then an inflammatory cytokine array assay was used to determine the secretion levels of inflammatory cytokines. In Fig. 4a, proinflammatory cytokines were presented as red color, antiinflammatory cytokines were presented as blue color. The tendency of anti-inflammatory cytokines was clear; the 60 min AIMS treatment + IFN-γ group had the highest concentration. The control + IFN-γ group also had higher concentrations than the control group, indicating that the IFN-γ pretreatment can stimulate MSCs to secrete more anti-inflammatory cytokines. The results for proinflammatory cytokines and chemokines are not very consistent. There were 16 targets that the IFN-γ pretreatment would decrease their secretion, including EOTAXIN, EOTAXIN-2, IFN-γ, IL-1β, IL-2, IL-3, IL-6sR, IL-7, IL-12p70, IL-17, IP-10, MIGMI, MIG, MIP-1β, MIP-1δ, and TNF-α. For these targets, the 60 min AIMS treatment group had the highest concentration, and the control group had the secondhighest concentration. After the IFN-γ pretreatment, these targets in the 60 min AIMS treatment + IFN-γ group were lower than the control group, and the control + IFN-γ group had the lowest concentration. There were also 9 targets with secretion profiles that the IFN-γ pretreatment did not change, including GCSF, GM-CSF, ICAM-1, I-309, IL-1α, IL-12p40, IL-15, MCP-1, and MCP-2. For these targets, both the 60 min AIMS treatment group and the 60 min AIMS treatment + IFN-γ group had higher concentrations than the control and control + IFN-γ groups. We also repeated the above-mentioned ELISA assay for the 8 targets to better observe the cytokine secretion profile. As shown in Fig. 4b–i, FGF-2, IGF-1, VEGF, HGF-1, and IL-10 showed a similar tendency. The 60 min AIMS treatment + IFN-γ group had the highest concentration. It was much higher than the control and control + IFN-γ groups. The 60 min AIMS treatment group was also higher than the control and control + IFN-γ groups. For IL-6, IFN-γ, and TNF-α, the 60 min AIMS treatment group had the highest concentration. The other three groups had similar profiles. According to our results shown in Fig. 2, the 60 min AIMS treatment group can improve the secretion of both pro- and anti-inflammatory factors. For anti-inflammatory factors and growth factors, the IFN-γ

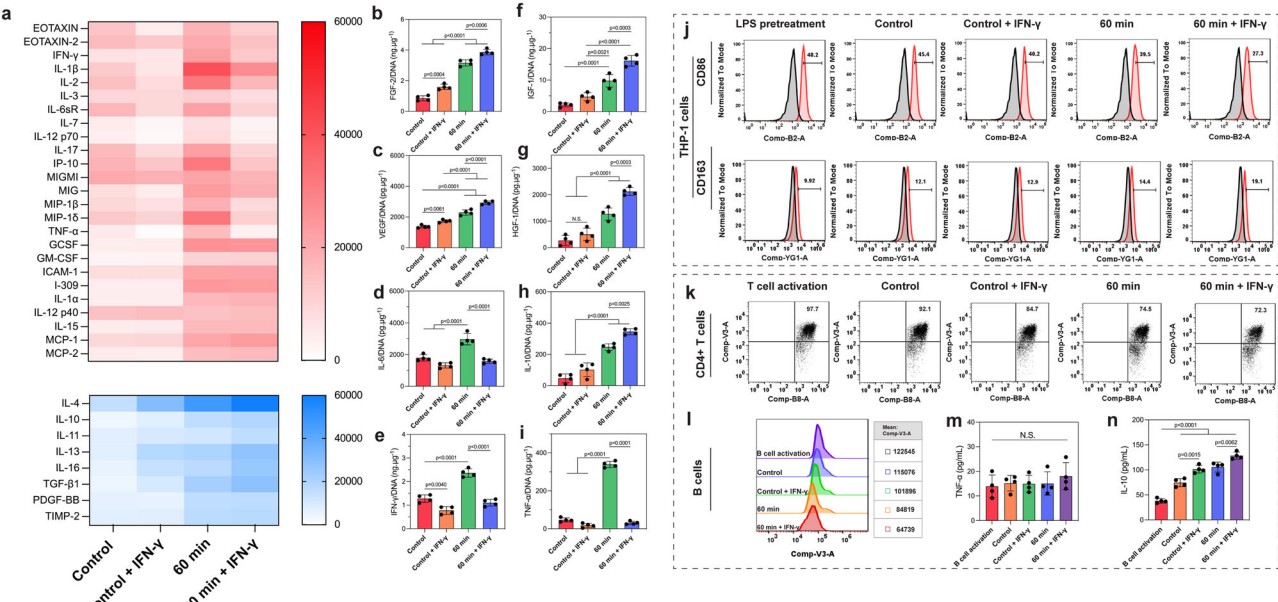

**Fig. 4 | Harnessing the AIMS-enhanced MSC secretome for immunomodulation applications. a** Heatmap measuring the inflammatory factors secreted from IFN-γ primed MSCs from different groups. **b–i** ELISA assay measuring the secretion of FGF-2, IGF-1, VEGF, HGF-1, IL-6, IL-10, IFN-γ, and TNF-α from different groups after incubating for 4 days. **j** Flow cytometry analysis measuring the percentage of CD86 and CD163 positive human THP-1 cells after incubating with the conditional medium from MSCs for 2 days. **k** Flow cytometry analysis measuring the percentage of CD4 positive T cells after incubating with the conditional medium from MSCs for 3 days. **l** Flow cytometry analysis measuring the B cell proliferation after incubating with the conditional medium from MSCs for 7 days. **m, n** ELISA assay measuring the secretion of TNF-α and IL-10 from B cells. The statistical analysis was performed using one-way ANOVA with Tukey's post-hoc test. Data are graphed as the mean ± SD (*n* = 4 tests, biological repeats). Source data are provided as a Source Data file.

pretreatment of MSCs can further improve their secretion. However, for pro-inflammatory factors and chemokines, some of the targets were not influenced by the IFN-γ pretreatment.

Next, the conditional medium from each group was collected and used for incubating with M1 macrophages, activated T cells, and activated B cells. Firstly, human THP-1 cells (human monocytes) were used in this study. After inducted by Phorbol-12-myristate 13-acetate (PMA), the cells were further polarized to the M1 (pro-inflammatory) phenotype with lipopolysaccharides (LPS) beforehand. LPS-treated cells were used as a positive control without adding an MSC-conditioned medium. MSCs with/without IFN-γ treatment were denoted as the 60 min AIMS treatment + IFN-γ group and the 60 min AIMS treatment group, respectively. First, the cell viability of THP-1 cells after incubation with the culture medium from different groups was studied. As shown in Supplementary Fig. 9, after 1 day, each group had no side effects on THP-1 cell viability. After 2 days, the cell viabilities of the control + IFN-γ group, the 60 min AIMS treatment group, and the 60 min AIMS treatment + IFN-γ group were even higher than the control group. These results indicated that the addition of MSC conditioned medium was beneficial for the viability of THP-1 cells, which could be attributed to the secretion of various cytokines from the MSCs. After that, the expression of nitric oxide, a typical pro-inflammatory indicator, was measured. As shown in Supplementary Fig. 10, after 1 and 2 days, the nitric oxide concentrations in the 60 min AIMS treatment group and the 60 min AIMS treatment + IFN-γ groups were statistically lower than that of the LPS-treated group and the control group. And they were lower than the control and control + IFN-γ groups as well. Next, flow cytometry assay was used to detect the phenotype switch of THP-1 cells (Fig. 4j). After being induced by PMA, all the cells expressed macrophage marker CD80. And we used CD86 and CD163 as the M1 and M2 phenotype markers. For CD86, the positive percentages of the LPS pretreatment, control, control + IFN-γ, 60 min AIMS treatment, and 60 min AIMS treatment + IFN-γ groups were 48.2%, 45.4%, 40.2%, 39.5%, and

27.3%, respectively. The number of CD86 positive cells gradually decreased. As for CD163, the positive percentages were 9.92%, 12.1%, 12.9%, 14.4%, and 19.1%, respectively. The number of CD163 positive cells increased. In other words, the 60 min AIMS treatment + IFN-γ group had the largest number of M2 phenotype macrophages and lowest number of M1 phenotype macrophages. The control + IFN-γ group also had a larger amount of M2 phenotype macrophages and less M1 phenotype macrophages compared to the control group, which meant that the IFN-γ pretreatment indeed was effective to induce the immunomodulative activities of MSCs.

Subsequently, we examined whether the enhanced MSC secretome can modulate the functions of T cells and B cells. For T cells, we used commercialized Human T-Activator CD3/CD28 beads to activate T cells. As shown in Fig. 4k and Supplementary Fig. 11, 97.7% of cells expressed CD4 marker and 97.8% of cells could express CD8 marker. Then the cells were incubated with the conditional medium of each group. The percentage of CD4 positive cells of control, control + IFN-γ, 60 min AIMS treatment, and 60 min AIMS treatment + IFN-γ groups were 92.1%, 84.7%, 74.5%, and 72.3%, respectively. The tendency was similar to CD86. However, for CD8, the differences among all groups were not significant. The 60 min AIMS treatment + IFN-γ groups could still keep 91.4% of CD8 positive cells. Therefore, the MSC secretome was more effective at the regulation of CD4 positive T cells than CD8 positive T cells. As for B cells, we used an antibody cocktail (F(ab)2 anti-IgM, IL-2, and anti-CD40 antibody) to stimulate original human B cells according to previous studies[56,57]. Then, the cells were incubated with the conditional medium of each group for 7 days. Following that, the cell proliferation and apoptosis were studied. As shown in Fig. 4l and Supplementary Fig. 12, for cell proliferation, compared with activated B cells, the cells in the conditional medium from the control and control + IFN-γ groups were slightly decreased. The 60 min AIMS treatment group was lower than the control and control + IFN-γ groups. The 60 min + IFN-γ group had the lowest number. These results indicate that the enhanced MSC secretome can effectively

inhibit activated B cell proliferation. However, for cell apoptosis, all the groups had no clear difference.

At last, we used ELISA assay to measure the cytokine secretion capacity of B cells from different groups. As shown in Fig. 4m and n, the secretion of TNF-α had no significant differences among all groups. However, for IL-10, the 60 min AIMS treatment + IFN-γ group had the highest concentration. All the control, control + IFN-γ, 60 min AIMS treatment, and 60 min AIMS treatment + IFN-γ groups had higher concentrations than the activated B cells without conditional medium incubation. In conclusion, although there was some inconsistency in our results—for example, some of the pro-inflammatory cytokines from MSCs still have comparable secretion level among all groups and the percentage of CD8 positive T cells and B cell apoptosis had no clear tendency—it is still reasonable to conclude that the MSC secretome enhanced via AIMS shows a more substantial immunomodulation effect than traditional MSC cultures.

## Discussion

It is well established that external mechanical forces and spatial arrangements can directly regulate the physiological behaviors of MSCs because of their high mechano-sensitivity[16]. External mechanical signals, such as ECM stiffness[58] and tensile force[59], can significantly affect the regulation of cytoskeleton rearrangement and mediate mechanically activated signal pathways that influence cell spreading, proliferation, and differentiation[60]. Previous studies have also demonstrated that the MSC secretome can be enhanced by improving the cell-matrix interaction and cell-cell interaction[22,23,61,62]. To investigate cell-matrix interactions, researchers have fabricated various ECM-mimic hydrogels to incubate with MSCs[21]. For example, grafting RGD peptide (bind integrins)[63], GFOGER peptide (collagen mimic peptide)[21], or other ECM mimic peptides onto polymer backbones has been used to culture MSCs. The focal adhesion, spreading, and migration of MSCs were found to be regulated by the peptide ligands.

Moreover, the secretome profile was also influenced due to the change in cell morphology[23]. To improve cell-cell interactions, MSCs were encapsulated in hydrogel droplets[64] or 3D cultured in microfluidic chambers[65]. Compared with the normal monolayer on cell culture dishes, the improved cell-cell interactions would up-regulate the expression of N-cadherin and further enhance the MSC secretome[23,26]. Undeniably, biomaterials and tissue engineering play an irreplaceable role in stem cell therapy, especially in vivo implantations. For example, material scaffolds can provide stable support and a gentle environment for MSCs to avoid the side effects of immune reactions or interferences from other biological components. MSCs can thus exert their functions effectively over the long term. However, for in vitro platform, using engineered biomaterials would introduce extra material components into the cell culture system, containing molecules that may interfere with cell growth and functionality. Additionally, collecting cells from surrounding materials to further test their secretome is complicated and time-consuming, which may also influence the result accuracy.

In this study, we developed the AIMS platform to enhance the MSC secretome. The AIMS platform has several unique advantages: first, unlike chemical methods, our platform has a negligible influence on normal cell functions, such as proliferation, intracellular calcium concentration, and ATP levels, which are comparable to monolayer cultures of MSCs. The aggregation of MSCs was mainly driven by the drag force from acoustic streaming, which was shown to be biocompatible with cells. Second, the AIMS platform can conveniently form 3D cell aggregates within the cell culture medium, and conventional cell culture substrates (e.g., petri dish, multi-well plates) can be used. The drag forces induced by acoustic streaming gently push the cells together in a non-contact manner to form the aggregates.

For the acoustic mechanism, in most of object manipulations by acoustic technologies, both the acoustic radiation force and drag force

from acoustic streaming play significant roles[66–68]. However, for our AIMS platform, it is the drag force that primarily facilitates cell concentration towards the device center. We provided visual aids in the form of Supplementary Fig. 13 to further clarify this. Supplementary Fig. 13a illustrates the distribution of the acoustic pressure amplitude, indicating a higher value compared to surrounding areas. The acoustic pressure amplitude creates an acoustic radiation force that radiates from the center towards the outer regions. Conversely, Supplementary Fig. 13b highlights the creation of acoustic streaming by these waves, where a low velocity region in the center is evidenced. The flow lines of acoustic streaming extend from the periphery to the center, assisting in the transportation of cells to the central area, consequently forming a rounded cell cluster. The interplay of these forces yields interesting results: the acoustic radiation force, due to its outward-pointing nature, resists cell concentration. Meanwhile, the drag force induced by acoustic streaming propels cells inward to the center. Crucially, our device operates at a low frequency of approximately 109 kHz. Under such conditions, the acoustic radiation force is substantially less impactful on cell movement compared to the drag force. Therefore, in our device, it is primarily the acoustic streaming mechanism that pushes cells concentrate to the center. As such, our AIMS platform can maintain cells in their culture environment and simplify collection for downstream biomedical studies.

We compared our AIMS platform with commercialized 3D cell culture microplates to distinguish the potential impacts from cell 3D culture and acoustofluidic stimulation. We found MSC proliferation in 3D microplate was inhibited, which was also supported by another study, which claimed that the formation of MSC spheroid would limit the oxygen diffusion, and the cell viability was affected even after 3 days[69]. Cell viability in our AIMS platform was not affected statistically, which may be attributed to the similar culture condition of AIMS with conventional cell culture substrates. It is also worth noting that the cells in 3D microplate and the cells in AIMS with different time durations showed distinct effects on secretome profile. Generally, 60 min of AIMS treatment had the best outcome. Cells in 3D microplate had comparable effects with 10 min in AIMS platform. We also found that in different groups, the morphologies of MSC aggregates were not identical, so the levels of cell–cell interactions may also be different. According to our results, the expression of N-cadherin varied among all groups, and the improved MSC secretome was strongly related to the functionality of N-cadherin. Compared with routine monolayer cultures, the formed 3D MSC aggregates have tight cell-cell interactions, and therefore, the expression of N-cadherin is up-regulated. The 3D microplate group also presented a higher level of N-cadherin expression than the monolayer cell culture, but it was still lower than AIMS treatment groups. The upregulation of N-cadherin is closely related to the recruitment of β-catenin and the canonical Wnt/β-catenin signaling pathway, which is responsible for various physiological functions of cells such as cell differentiation and paracrine signaling[50,51,69]. Our results show trends that agree with prior studies[23,26,50] on secretome enhancement, as the expression of N-cadherin has been shown to modulate the MSC final secretome products. The results are reasonable because it has been shown that acoustic streaming has played a similar role in hydrogel systems, which serve to confine and grow cells to improve their cell–cell interactions mechanically.

We also showed that the secretome profile of MSCs can be modified for different therapeutic uses. It is well accepted that after they are primed by IFN-γ, MSCs will enter an immunomodulatory state[55,70]. According to our results, the secretome profile after AIMS treatment had an overall improvement, including growth factors, pro-inflammatory factors, and anti-inflammatory factors. Other studies also reported that the paracrine effect of MSCs was enhanced in a full scale[71,72]. Therefore, using IFN-γ to pretreat MSCs and help them enter the immunomodulatory state was essential for the therapeutic

practice of MSCs. After being primed by IFN-γ, we found the secretion of anti-inflammatory factors was further improved. The primed MSCs in our AIMS platform had the highest concentrations of anti-inflammatory factors. Although there were some inconsistences for the secretion of pro-inflammatory factors and chemokines, the collected secretome was still efficient to inhibit the pro-inflammatory response of M1 phenotype macrophages, suppress T cell activation, and support B cell functions. It was also interesting to note that the collected secretome was more effective to regulate CD4[+] T cells rather than CD8[+] T cells. Moreover, the secretome was more effective to inhibit B cell proliferation rather than enhance apoptosis. More studies need to be done in the future to explore the underlying mechanism. Overall, MSC aggregates showed the most potent anti-inflammatory response, which holds great promise in developing future stem cell therapies. Our results indicate that AIMS represents a promising solution to confine MSCs in vitro and tune their immunomodulatory properties in a simple, non-contact manner. The improved secretome could be a good alternative to stem cell therapy, as potential risks associated with stem cell transplantation, such as immune rejection, tumorigenicity, or unwanted differentiation, are significantly reduced or eliminated. The production of conditioned medium can be easily scaled up, ensuring a consistent and standardized treatment option, which may largely avoid the shortage of stem cell supply in clinic. The improved secretome can also be integrated with biomaterials for implantation to support in situ tissue regeneration consistently and prevent the early degradation of proteins.

## Methods

### Ethical statement

Human derived mesenchymal stem cell line used in this study was provided by commercial source (Zen-bio). All donor identifiers were removed prior to receiving the cells.

**Device design and working principle.** To design the device, a piezoelectric ring (SMR1357T12R412WL, Steiner & Martins, Inc., USA) was tightly attached to the bottom of glass cell culture dishes (801002, NEST Biotechnology, diameter: 15 mm) by using a thin epoxy layer (Permatex, USA). The inner and outer diameter of the piezoelectric ring was 7 and 13.5 mm, respectively. To generate a circular standing flexural wave, excitation signals from a function generator (SDG1050, Siglent, Germany) were applied to the piezoelectric ring at its resonant frequency (108 kHz) at a driving voltage of 6 $V_{pp}$. A 200 μL liquid droplet of suspended cells was placed at the center of the glass bottom, where acoustic waves were transmitted through the glass substrates and into the cell suspension. The flexural acoustic waves generated an inward acoustic streaming effect, which was used to aggregate cells toward the center of the droplet.

Simulation was performed to visualize the acoustic intensity distribution and acoustic streaming effect conducted in COMSOL Multiphysics 5.6 (Supplementary Fig. 14). A fully coupled three-dimensional model was developed by coupling the acoustic pressure module and solid mechanics module to calculate the acoustic intensity distribution. In brief, the model setup consisted of a ring piezoelectric transducer (PZT) located at the base of a glass substrate, affixed with a thin layer of Epoxy. Positioned atop this glass substrate was a small droplet. The table below delineated the geometric parameters of our model:Table 1

For the calculation of the acoustic pressure distribution, we coupled the Electrostatics, Solid Mechanics, and Thermoviscous Acoustics

modules using default interfaces in Comsol Multiphysics 5.6. We assigned a ground terminal to the upper surface of the PZT, whereas the lower surface received an electric potential of 30 V. The droplet's outer surface was treated with a slip impedance of air. This part of the study was conducted in the frequency domain at 109.2 kHz. In the subsequent phase of the study, intended to compute acoustic streaming, we applied the body force[73] derived from the preceding acoustic pressure analysis to the Laminar Flow module. To aid module convergence, we constrained this module with a pressure point of 0 Pa on the bottom surface.

**Cell culture.** Cryopreserved human bone marrow-derived mesenchymal stem cells (MSCs) were obtained from Zenbio (HBMMSC-F). MSCs were cultured in Bone Marrow Stem Cell Growth Medium (Zenbio, BMSC-1). The medium was replaced on the first day and then every 2 days thereafter. After reaching 90% confluence, cells were digested with TrypLE™ (Gibco, 12604-013) for sub-culture. Cells in the third passage were used in this study. Human THP-1 cells were bought from Sigma-Aldrich (88081201), and cultured in RPMI 1640 medium (Gibco, 72400120) with 10% of FBS (v/v). Phorbol-12-myristate 13-acetate (PMA) (Sigma-Aldrich, 100 nM) was used to differentiate THP-1 cells to macrophages. Human Normal Peripheral Blood Mononuclear Cells (PBMCs) were supplied by Zenbio (SER-PBMC-F) and cultured in RPMI 1640 medium with 10% of FBS (v/v). For PBMC activation, the Human T-Activator CD3/CD28 beads (11131D) were used according to the manufacturer's instructions. Purified human B Cells were obtained from IQbiosciences (IQB-Hu1-B5) and cultured in RPMI 1640 medium with 10% of FBS (v/v). For B cell activation, a stimulation cocktail (10 mg/ml F(ab)2 anti-IgM, $10^3$ IU IL-2, and 5 mg/ml anti-CD40 antibody) was used according to a previous study[57]. After 7 days of culture, the activated B cells were ready for further use.

**Cell sample preparation.** Similar to a previous study, the glass substrate of the device was pre-treated with Polyvinyl alcohol (Mowiol® 18-88, Sigma-Aldrich Chemical. Co.) to prevent early adhesion of MSCs to the substrate[74]. Briefly, Mowiol® 18-88 was dissolved in PBS (1 wt.%), and 500 μL of the solution was added to the substrate and incubated at 37 °C for 1 h. After that, the solution was discarded, and the substrate was washed with PBS three times. The device was then dried in a drying oven and sterilized by UV irradiation. For the following MSC cell experiments, a 200 μL droplet of suspended cells (1 ×10⁴ cells) was added to each device in a sterile environment. After sealing the lid of the glass culture dishes, the device was connected to a function generator. After turning the acoustic device on, the samples were treated for 10, 30, and 60 min, respectively. A cell suspension directly placed onto the substrate without AIMS assembly served as the control group. For the 3D Microplate-cultured samples, the cell droplets with the same volume were added into the commercialized Corning® Spheroid Microplates (10185), and the sample was denoted as 3D Microplate.

**Characterization of MSC aggregates.** To observe the cell aggregation of different groups, MSCs were labeled with calcein-AM (Invitrogen, Life Technologies, MA, USA) beforehand. Cell aggregation was then observed and imaged using Nikon imaging software (NIS-Advanced, Nikon, Japan) with a CCD digital camera (CoolSNAP HQ2, Photometrics, Tucson, AZ, USA).

To test the MSC proliferation, a CyQUANT™ Cell Proliferation Assay (Invitrogen, C7026) was performed. The dishes without Mowiol® 18-88 pretreatment were a positive control group (TCPs). The samples were further incubated at 1 and 4 days, and the total DNA content of each group was measured according to the manufacturer's instructions using a spectrophotometric microplate reader (bioTek Instruments, Inc.)

For the morphology observation, the samples were fixed with 4% paraformaldehyde for 15 min at 4 °C and permeabilized with 0.2%

**Table 1 | Geometric parameters of our model (all units in mm)**

| $R_1$ | $R_2$ | $R_3$ | $h_1$ | $h_2$ | $h_3$ | $h_4$ | d |
|---|---|---|---|---|---|---|---|
| 7 | 9 | 3.22 | 1.2 | 0.2 | 0.2 | 0.9 | 6.5 |

Triton to observe the morphology of the MSC aggregates X-100 for 2 min. The samples were incubated with Alexa Fluor™ 555 Phalloidin (Invitrogen, A34055, 1:400) at 4 °C overnight and incubated with Hoechst 33342 (Invitrogen, H3570) for 5 min. After being washed with PBS three times, the samples were observed and imaged.

**Secretome studies of MSC aggregates.** The secretion of FGF-2, IGF-1, VEGF, HGF-1, IL-6, IL-10, IFN-γ, and TNF- α of the MSC aggregates after incubation for 4 days was tested by the ELISA kits (Abcam, ab246531, ab222510, ab275901, ab108873, ab178013, ab185986, ab174443, and ab181421) according to the manufacturer's instructions. A broader secretome profile was studied using a cytokine array assay. After 4 days of incubation, the supernatant of four biological repeats of each group was collected and incubated using the cytokine assay (Abcam, ab133998) according to the manufacturer's instructions. The results of ELISA and the cytokine array assay were normalized by the DNA content of each group.

Similarly, supernatant from MSC aggregates incubated for 4 days was collected, and the total number of exosomes was obtained using a Total Exosome Isolation Reagent (Invitrogen, 4478359). Briefly, the supernatant was centrifuged at $2000 \times g$ for 30 min at 4 °C to remove cells and debris. Then the reagent was added to the supernatant at a 1:2 reagent: supernatant ratio. After thorough mixing, the samples were incubated at 4 °C overnight and then centrifuged at $10,000 \times g$ for 60 min at 4 °C. The supernatant was discarded, and the exosomes at the bottom were extracted.

To calculate the number of secretome-produced exosomes in each group, an EXOCET Exosome Quantitation Kit (SBI, EXOCET96A-1) was used according to the manufacturer's instructions. The exosome abundance of each group was also verified by directly measuring the acetylcholinesterase (AchE) activity (Abcam ab138871) according to the manufacturer's instructions. The size distributions of the exosomes from each group were measured using a nanoparticle tracking analysis (NTA, Nanosight LM10, Malvern, England) system. The number of exosomes from each group was also verified using NTA analysis.

**Characterization of the enhanced MSC secretome.** The expression of N-cadherin was examined using Western blotting, Q-PCR, and flow cytometry. The samples were collected after 3 days of incubation. For Western blotting assay, the anti-human CD325 (N-cadherin) Antibody (Biolegend, 350802, 1:200) was used as the primary antibody, and the Goat Anti-Mouse IgG H&L (HRP, Abcam, ab205719, 1:10,000) was used as the secondary antibody. For Q-PCR assay, the Taqman® kit for CDH2 gene was used (Thermofisher, 4400291). The Actin gene was used as the reference. For flow cytometry assay, the PE anti-human CD325 (N-cadherin) Antibody (Biolegend, 350805, 1: 20) was used to incubate with the samples for 30 min. Following incubation, the cells were detected using flow cytometry (BD, Bioscience, USA). Cytoimmuno-fluorescence staining was also used to visualize the N-cadherin expression in Control, 10 min, 30 min, and 60 min group. After 3 days of incubation, the samples from each group were treated with 4% paraformaldehyde at 4 °C for 30 min. They were then permeabilized with 0.2% Triton X-100 for 5 min and blocked with a 1% BSA solution for 1 h. The samples were then incubated with primary Anti-N-cadherin antibody (Abcam, ab245117, 1:500) and Alexa Fluor™ 555 Phalloidin (1: 400) at 4 °C overnight. Alexa Fluor-488 secondary antibody (Abcam, ab150077, 1: 1000) was used to stain the samples for another 30 min. Finally, cell nuclei were stained by Hoechst 33342 for 10 min. Immunofluorescence images were collected with Nikon Imaging Software (NISAdvanced, Nikon, Japan) with a CCD digital camera (CoolSNAPHQ2, Photometrics, Tucson, AZ, USA).

To investigate whether the MSC secretome was enhanced due to the upregulation of N-cadherin, an N-cadherin neutralizing antibody (GC-4, Sigma-Aldrich) was used. Before preparing cell samples, MSCs were pre-treated with a neutralizing antibody solution (50 µg/mL, in PBS) for 1 h. After AIMS assembly, the cytokine array (Abcam, ab133998) assay and ELISA assay were performed as mentioned above.

To measure the intracellular $Ca^{2+}$ concentration, samples were washed with PBS three times after AIMS assembly (or further incubated for 4 days). A Fluo-4 NW Calcium Assay Kit (Invitrogen, F36205) was used to measure the $Ca^{2+}$ influx of each group according to the manufacturer's instructions.

To measure the ATP level, after AIMS assembly of MSCs, the samples were gently washed with PBS three times. The ATP level of each group was then measured using an ATP Assay Kit (Abcam, ab83355) according to the manufacturer's instructions.

**Immunomodulation of the MSC secretome.** To modify the secretome of the MSCs into an immunomodulatory phenotype, the MSCs were licensed by IFN-γ. In brief, 20 ng/mL of IFN-γ (Sigma-Aldrich, I17001, dissolved in culture medium) was incubated with MSCs for 12 h. Cells were then collected and 60 min of AIMS assembly was performed to aggregate the cells. The cell aggregates were then incubated for 4 days, and the conditioned medium of each group was collected. To determine the cytokine profile of the immunomodulated secretome, the supernatant of four repeats from each group was collected and incubated with an inflammatory cytokine array assay (Abcam, ab134003) according to the manufacturer's instructions. The ELISA assays were conducted as above mentioned to verify the cytokine secretion.

Next, the immuno-modulatory effects on different immune cells were performed. For human THP-1 cells, 100 ng/mL of lipo-polysaccharide (LPS, Sigma-Aldrich, L2018, dissolved in culture medium) was incubated with cells for 12 h to induce the M1 polarization. For the following immunomodulation experiments, the conditioned medium was mixed with RPMI 1640 medium (1:1, v/v) to culture the LPS-treated THP-1 cells. The CCK-8 Assay was conducted to measure cell proliferation of the THP-1 cells. To test the nitric oxide (NO) production, the culture medium was collected after 2 days of incubation. The NO concentration was measured using a Nitric Oxide Assay Kit (Abcam, ab65328) according to the manufacturer's instructions. To examine the phenotype switch of THP-1 cells, APC anti-human CD68 Antibody (Biolegend, 333809, 1:20), FITC Anti-CD86 antibody (Abcam, ab77276, 1:20), and PE Anti-CD163 antibody (Abcam, ab95613, 1:20) were used for flow cytometry (BD, Bioscience, USA) after incubating for 2 days.

For human T cells, the conditioned medium was mixed with RPMI 1640 medium (1:1, v/v) to culture the activated PBMCs. Then after 3 days, PerCP anti-human CD4 antibody (Biolegend, 300527, 1:20), APC anti-human CD8a antibody (Biolegend, 301014, 1:20), and Pacific Blue™ anti-human CD25 antibody (Biolegend, 356129, 1:20) were used for flow cytometry to evaluate the cell activation.

For human B cells, the conditioned medium was mixed with RPMI 1640 medium (1:1, v/v) to culture the activated B cells. After 7 days, the CellTrace™ Violet Cell Proliferation Kit (Invitrogen, C34571) and PE Annexin V Apoptosis Detection Kit I (BD, 559763) were used for flow cytometry to evaluate the cell apoptosis and proliferation. Also, the secretion of TNF-α and IL-10 was tested by ELISA assay. B cells were incubated with the conditioned medium for 3 days. Then the medium was changed to RPMI 1640 medium and incubated for further 3 days.

### Statistical analysis
The statistical analysis was performed using one-way ANOVA with Tukey's post-hoc test by GraphPad Prism 9.0. Images are processed by ImageJ (NIH, USA). The data were presented as a mean ± standard deviation (SD) of multiple biological replicates (as indicated in the figure legends). N.S.: not significant.

## Reporting summary

Further information on research design is available in the Nature Portfolio Reporting Summary linked to this article.

## Data availability

The data supporting the figures are provided with this paper as Source data file. The datasets are available from the corresponding author (T.J.H., email: tony.huang@duke.edu), and the data requests will be fulfilled within two weeks. Source data are provided with this paper.

## Code availability

The acoustic wave simulations were performed with commercial software COMSOL Multiphysics 5.6. No special codes were developed for this study. The computation details are described in the Methods section and are available from the corresponding author (T.J.H., email: tony.huang@duke.edu), and the requests will be fulfilled within two weeks.

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

## Acknowledgements

This study was partly funded by the National Institutes of Health (R33CA223908 (T.J.H.), R01GM141055 (T.J.H.), and R01GM145960 (L.P.L.)) and National Science Foundation (ECCS-1807601 (T.J.H.)).

## Author contributions

Y.H. led the experimental work. S.Y., L.P.L., and T.J.H. provided guidance and contributed to the experimental design. Y.H. and P.L. developed the system concept and contributed to the device fabrication and improvements. K.J., J.Z., K.L., Z.M., and R.Z. contributed to biological experiment design and data analysis. R.B. contributed to data analysis and manuscript improvement. J.X. did the theoretical modeling and numerical simulation. C.L. contributed to data analysis. Y.H., S.Y., L.P.L, and T.J.H. co-wrote the manuscript.

## Competing interests

T.J.H. has co-founded a start-up company, Ascent Bio-Nano Technologies Inc., to commercialize technologies involving acoustofluidics and acoustic tweezers. All remaining authors declare no competing interests.
