## [Peer Review File · Nature Communications]

REVIEWER COMMENTS

Reviewer #1 (Remarks to the Author):

The MSC secretome has become an attractive subject of investigation due to its role in immunomodulation, anti-inflammatory properties, angiogenesis and anti-apoptosis.

In this manuscript AIMS technology is established to improve MSC secretome with high efficiency in terms of concentration of exosomes released and specific cytokines secretion. This opens perspectives for personalized secretome-based therapies.

This is a very interesting manuscript that significantly advance knowledge on the subject.

In addition to macrophage polarization a revised version of the manuscript should provide additional biological evidence of improved effect of AIMS generated secretome in a system such as T and B lymphocytes.

I would suggest to cite in the Introduction more recently published references on the sybject of MSC and their secretome application i.e PMID 24268069, 27356536, 26201487.

Reviewer #2 (Remarks to the Author):

Dear Author, Dear Editor,

Thank you for the opportunity to share your article entitled "Acoustofluidic Interfaces for the Mechanobiological Secretome of MSCs: AIMS".

In the mentioned study, you present an acoustofluidic strategy to enhance the secretion of trophic factors of MSCs. You show that the developed AIMS platform promotes the formation of 3D aggregates and detected higher concentrations of various growth factors and cytokines in the conditioned medium of 3D aggregates than in the control group (MSC cultures without acoustofluidic stimulation). You attribute the increased cytokine levels in the supernatant of the 3D aggregates to enhanced cell-cell interactions mediated by N-cadherins. Furthermore, you provide evidence that the increased levels of molecular factors in the supernatant of the 3D aggregates have immunomodulatory effects on a murine macrophage cell line (RAW 264.7 cells), which in turn reduces their pro-inflammatory secretion profile (after LPS stimulation). This effect could be enhanced by pre-stimulation of MSCs with IFN- γ . In summary, this study confirms the well-established concepts that the secretory properties of MSCs are significantly affected by the amount of cell-cell contact and that stimulation of cells with IFN- γ increases their immunomodulatory phenotype. For the most part, the quality of the data is technically sound and presented in sufficient detail; however, the novelty value of the results is limited and advances our understanding of the mechanisms underlying the secretory (or immunomodulatory) properties of MSCs only to a limited extent. Also, in terms of potential therapeutic application of the results, fear to say that your study provides little evidence to advance the field. The use of the AIMS platform for basic research questions is very promising, but mainly of interest to a specialized audience.

Specific main comments:

1.) The formation of MSC aggregates with the AIMS platform is very interesting, but it is important to introduce further controls to classify the results. MSCs also form 3D spheroids in ultra-low attachment plates, which would allow discrimination between the influence of 3D structure and acoustofluidic stimulation. An alternative control (which is mandatory) is MSCs cultured on polyvinyl alcohol treated plates but not acoustofluidically stimulated.

2) The differences in cytokine/growth factor concentrations in the conditioned supernatant of the different cultures are very interesting. However, to what extent can you rule out that the observed

differences are not due to differences in cell number? You write, "The cells in all groups showed a good proliferation profile, and no short-term adverse effects were observed in the 30- and 60-minute groups. The assay used for this purpose (CCK-8) is suitable for adherent monolayer cells, but not for 3D spheroids. Studies by other colleagues have already shown that the assay is inappropriate for the determination of cell number in 3D spheroids. More direct methods such as DNA content determination (e.g., using CyQuant proliferation assays) or direct cell counting after dissociation of the 3D structure are mandatory to determine the exact cell number (see <https://doi.org/10.1016/j.jtcms.2020.09.004> - especially Figures 3 and 4). Figures 3b (FDA staining) and 3c (Hoechst) clearly show the differences in cell number between cultures within the first 2 days.

- 3) The secretion profiles and exosome diameter profiles are very different between the acoustofluidically stimulated groups. Is there an explanation for this? Does this have a functional impact (e.g., on immunomodulatory effects?).
- 4) There is a lack of critical discussion of the secretion data - known proinflammatory factors (TNF- α , IFN- γ , IL-6) have much higher levels in the AIMS cultures than in the controls - how can this be reconciled with immunomodulatory effects?
- 5) The qualitative differences in the amount of N-cadherin are remarkable. Could you please also present the quantitative differences at mRNA and protein level normalized to the one housekeeping gene/protein or cell number to make the interpretation of the results comprehensible.
- 6) After functional blocking of N-cadherin, the secretion profiles of the different groups change but are not identical. Is it correct that the control was also treated with N-cadherin antibodies? Could you please add corresponding reference samples (e.g. 60 min and control without N-cadherin blocking) to the heatmap?
- 7) What is the impact of functional blocking of N-cadherin on the immunomodulatory properties of the conditioned medium? Does it also affect IFN- γ licensing of the MSCs?
- 8) In the studies on immunomodulatory effects, control monolayer cultures with IFN- γ stimulation are missing. Without this control, evaluation and interpretation regarding the influence of AIMS are difficult. In other words, IFN- γ stimulation could in any case be sufficient to produce a comparable effect in monolayers without AIMS.
- 9) The presented heatmap (figure 4b) seems to contradict in essential parts the heatmap in figure 2b? The analytes IL-13, TNF- α , TNF- β , TGF- β , RANTES, MCP-1, IP-10, IL-16, IL-12 p40/p70, IL-3, IFN- γ are significantly higher in the 60min group than in the control in figure 2b, while it is exactly the opposite in figure 4b? IL-10 is comparable between 60min group and control in Figure 2b and not different as in Figure 4b. Finally, MIP-1 β in Figure 2b is higher in the control than in the 60 min group, while in Figure 4b it is the other way around. In addition to the heatmaps in Figure 2, 3 and 4, please show quantitative (ELISA / multiplex ELISA) data for selected (but the same) cytokines as surrogate markers for the secretome. I would suggest based on your reasoning: TNF- α , IFN- γ , IL-6, IL-10, FGF-2, VEGF, and HGF. This would make the figures comparable to each other and better justify your argumentation.
- 10) Regarding 4a: Why should anti-inflammatory therapy be used to treat an infection?
- 11) Why did you use a murine cell line to study the effects of conditioned medium from human cells?
- 12) The qualitative assessment of macrophage polarization by CD86 is good, however, 2b macrophages also express CD86 - does the conditioned medium of MSCs change the macrophage polarization from M1 to M0 or to M2. Please try to generate quantitative data and use a larger panel (e.g. MHCII, CD68, CD80, CD86, CD163) to allow classification of results.
- 13.) Please indicate if the N numbers in the captions are biological or technical replicates. MSCs

from how many donors were used? - could you provide age and sex information?

14.) The statistical analysis is not sufficient. Was the Student's t-test two-sided? Student's t-test is only permissive for comparisons of two groups. For comparisons of more than two (independent) groups, it is mandatory to adjust the alpha levels and apply an appropriate statistical test with appropriate correction procedures for the p-value for multiple comparisons.

15) Please try to have a more balanced discussion in relation to biomaterials. For clinical application of cell therapy, biomaterials are often necessary to allow transplantation of cells in the first place and to keep them at the site of injury (carrier function), furthermore biomaterials protect regenerative cells from negative influences of local immune cells (support function). Finally, many MSC isolations express tissue factor, which precludes systemic administration of the cells, as this can be a high risk for blood-mediated inflammatory reactions (IBMIR).

16) Please also discuss the potential application of your results in terms of clinical and basic research. What would be the therapeutic advantage of using conditioned medium instead of cell therapy? Proteins have a much shorter half-life than living cells? Biomaterials have the advantage that they can stimulate or inhibit specific mechanisms of cells in 3D.

Reviewer #3 (Remarks to the Author):

This paper applies a method based on acoustic trapping/streaming to generating 3D cultures of MSCs and harvesting the excreted proteins and exosomes (secretome). The novel aspect is that the effect of acoustofluidic culture on the composition of the secretome has been investigated. The authors suggest that this composition may be influenced by culture in the acoustic trap. The paper is timely, proposes an interesting device and concept, and provides a very interesting characterisation of the excreted proteins and exosomes.

In the paper the phrase improved secretome is used often but not well defined. Supposedly the authors mean a secretome with a composition that would be more effective if used therapeutically? If this is the claim, a clearer definition of what this means in terms of specific molecules, or for instance over-all concentration levels, needs to be supplied or discussed in more detail. Alternatively, a more nuanced way of talking about the changes to the secretome may be sufficient.

Key aspect of assessing the originality is to see what effects on the secretome that can be attributed to the acoustic trapping/streaming (which may add for instance mechanical stimulation and mixing) and what is the general effect of 3D culture. For instance, reference 27 seems to suggest that large effects on the secretome can be found in a generic 3D culture. Would there be a significant difference when comparing the method to for instance 3D culture in a well plate with conical bottom, a hanging droplet culture or similar? All of the controls in the paper are made to a monolayer culture so it is hard to say whether the improvements are general 3D culture features or specific to the acoustic trap. If the enhancement of the secretome could be accomplished on any 3D culture platform those may be significantly easier to scale for parallel operation which would be needed if this was to be used therapeutically. On the other hand, if the effects on the secretome was specific to the acoustic setup that would be highly interesting. To evaluate this, I think additional experiments may be needed.

Suggested experiments:

1. A comparison with conventional spheroid culture may make it possible to assess that the effects of the AIMS method beyond what is accomplished in any system providing 3D aggregation.
2. It would be interesting to further explore a wider range of actuation times. The paper describes how the cells are aggregated within 90 s. A comparison between turning the actuation off directly after aggregation to having the aggregation on for an entire assay (days) might also provide interesting results for analysing the effect of the acoustic forces.

The acoustofluidic device presented is very interesting in itself, with a minimalistic and straight-

forward configuration. It is not clear however if the mechanism for particle manipulation is based on acoustic streaming or radiation forces. In the paper the authors seem to suggest that acoustic streaming is the primary mechanism, however, in a previous publication:

(1) Oberti, S.; Neild, A.; Dual, J. Manipulation of Micrometer Sized Particles within a Micromachined Fluidic Device to Form Two-Dimensional Patterns Using Ultrasound. *J. Acoust. Soc. Am.* 2007, 121 (2), 778–785. <https://doi.org/10.1121/1.2404920>.

A similar configuration where surface modes on a plate are also used to manipulate particles radiation forces are more important. It would be interesting if the authors could elaborate on why streaming is the most important mechanism or whether it is a combination effect.

I found very little discussions and citations to the prior literature concerning acoustic trapping and formation of 3D cultures. Very much has been done on this topic and in addition acoustic forces has been used to form 3D cultures of MSCs specifically (Jeger-Madiot et al.). Some relevant references on this topic could include:

Pioneering work:

(1) Liu, J.; Kuznetsova, L. A.; Edwards, G. O.; Xu, J.; Ma, M.; Purcell, W. M.; Jackson, S. K.; Coakley, W. T. Functional Three-Dimensional HepG2 Aggregate Cultures Generated from an Ultrasound Trap: Comparison with HepG2 Spheroids. *J. Cell. Biochem.* 2007, 102 (5), 1180–1189. <https://doi.org/10.1002/jcb.21345>.

A review:

(2) Olofsson, K.; Hammarström, B.; Wiklund, M. Ultrasonic Based Tissue Modelling and Engineering. *Micromachines* 2018, 9 (11), 594. <https://doi.org/10.3390/mi9110594>.

Recent advances/devices:

(3) Jeger-Madiot, N.; Arakelian, L.; Setterblad, N.; Bruneval, P.; Hoyos, M.; Larghero, J.; Aider, J. L. Self-Organization and Culture of Mesenchymal Stem Cell Spheroids in Acoustic Levitation. *Sci. Rep.* 2021, 11 (1), 1–8. <https://doi.org/10.1038/s41598-021-87459-6>.

(4) Luo, Y.; Gao, H.; Zhou, M.; Xiao, L.; Xu, T.; Zhang, X. Integrated Acoustic Chip for Culturing 3D Cell Arrays. *ACS Sensors* 2022, 7 (9), 2654–2660. <https://doi.org/10.1021/acssensors.2c01103>.

(5) Chen, K.; Wu, M.; Guo, F.; Li, P.; Chan, C. Y.; Mao, Z.; Li, S.; Ren, L.; Zhang, R.; Huang, T. J. Rapid Formation of Size-Controllable Multicellular Spheroids: Via 3D Acoustic Tweezers. *Lab Chip* 2016, 16 (14), 2636–2643. <https://doi.org/10.1039/c6lc00444j>.

(5) Olofsson, K.; Carannante, V.; Ohlin, M.; Frisk, T.; Kushiro, K.; Takai, M.; Lundqvist, A.; Önfelt, B.; Wiklund, M. Acoustic Formation of Multicellular Tumor Spheroids Enabling On-Chip Functional and Structural Imaging. *Lab Chip* 2018, 18 (16), 2466–2476. <https://doi.org/10.1039/C8LC00537K>.

I would like to see a discussion about these methods in the paper and how the proposed design is different and perhaps advantageous for this particular application.

Figure 4c the viability is presented on the scale of 0-12 and not as a percentage of the population please clarify this.

The simulations presented here are not described in sufficient detail in the method section. Please provide information of which boundary conditions was used, and how the excitation was done such that they can be replicated. For this the details of the geometry and dimensions are also needed.

The heatmap in figure 2b shows that some of parts of the secretome is down regulated when comparing the 30 min to the 60 min conditions. This links to the discussion I would like to see about what constitutes an improved secretome. Are these proteins less important for the quality/function of the secretome?

I would like to say I'm not able to critically review all aspects of the in-depth immunology presented in this paper. This concerns primarily figure 4 and the selection of which proteins that are important to include in the panels and whether or not the pro-inflammatory macrophages is the most relevant choice for testing the secretome.

RESPONSE TO REVIEWERS' COCMMENTS

Reviewer 1

The MSC secretome has become an attractive subject of investigation due to its role in immunomodulation, anti-inflammatory properties, angiogenesis and anti-apoptosis. In this manuscript AIMS technology is established to improve MSC secretome with high efficiency in terms of concentration of exosomes released and specific cytokines secretion. This opens perspectives for personalized secretome-based therapies. This is a very interesting manuscript that significantly advance knowledge on the subject.

Response:

We sincerely thank the reviewer for the positive comments and constructive feedback. We have addressed each of your comments, adding details regarding the validation of our method, and we have revised the manuscript accordingly.

Comment 1:

In addition to macrophage polarization a revised version of the manuscript should provide additional biological evidence of improved effect of AIMS generated secretome in a system such as T and B lymphocytes.

Response 1:

Thanks for your helpful advice. Based on your comments, we added T and B lymphocytes-related experiments. In brief, we activated B and T cells and incubated them with the collected conditional medium. Then the percentages of CD4+ and CD8+ T cells were detected by flow cytometry assay. The cell proliferation and apoptosis of B cells were also measured by flow cytometry assay. The secretion of TNF- α and IL-10 from B cells was examined by ELISA assay. The results were organized in Fig. 4, Fig. S11, and Fig. S12 as below:

Figure 4. **k**, Flow cytometry analysis measuring the percentage of CD4 positive T cells after incubating with the conditional medium from MSCs for 3 days. **l**, Flow cytometry analysis measuring the B cell proliferation after incubating with the conditional medium from MSCs for 7 days. **m& n**, ELISA assay measuring the secretion of TNF- α and IL-10 from B cells. Data are graphed as the mean \pm SD (n=4), *p < 0.05, **p < 0.01, ***p < 0.001, ****p < 0.0001.

Figure S11 | Flow cytometry analysis measuring the percentage of CD8 positive T cells after incubating with the conditional medium from MSCs for 3 days.

Figure S12 | Flow cytometry analysis measuring the B cell apoptosis after incubating with the conditional medium from MSCs for 7 days.

According to our results, the percentage of CD4 positive cells of control, control + IFN- γ , 60 min, and 60 min + IFN- γ groups were 92.1, 84.7, 74.5, and 72.3%, respectively. However, for CD8, the differences among all groups were not very significant. The 60 min + IFN- γ group could still keep 91.4% of CD8 positive cells. Therefore, the MSC secretome was more effective for the regulation of CD4 positive T cells compared to CD8 positive T cells. As for B cells, for cell proliferation, compared with activated B cells, the cells in the conditional medium from the control and control + IFN- γ groups were slightly decreased. The 60 min group was lower than the control and control + IFN- γ groups, and the 60 min + IFN- γ group had the lowest percentage. These results indicate that the enhanced MSC secretome can effectively inhibit activated B cell proliferation. However, for cell apoptosis, there was no clear difference in the groups. The secretion of TNF- α also resulted in no significant differences among all groups. However, for IL-10, the 60 min + IFN- γ group had the highest concentration. All groups had higher concentrations than the activated B cells without conditional medium incubation ($p < 0.0001$). In conclusion, although some inconsistencies are present in the results - for example, some of the pro-inflammatory cytokines from MSCs still have comparable secretion level among all groups and the percentage of CD8 positive T cells and B cell apoptosis had no clear tendency - it is still reasonable to conclude that the MSC secretome enhanced via acoustofluidic assembly shows a more substantial immunomodulation effect than traditional MSC cultures.

Per the reviewer's comments, we have added the following information to the revised manuscript:

Changes to the manuscript (Page 13 in the main text):

“Subsequently, we also examined whether the enhanced MSC secretome can modulate the functions of T cells and B cells. For T cells, we used commercialized Human T-Activator CD3/CD28 beads to activate T cells. As shown in Fig. 4k and Fig. S11, 97.7% of cells expressed the CD4 marker and 97.8% of cells expressed the CD8 marker. The cells were then incubated with the conditional medium of each group. The percentage of CD4 positive cells of control, control + IFN- γ , 60 min and 60 min + IFN- γ groups were 92.1, 84.7, 74.5, and 72.3%, respectively. This tendency was

similar to CD86. However, for CD8, the differences among all groups were not very significant. The 60 min + IFN- γ groups could still keep 91.4% of CD8 positive cells. Therefore, the MSC secretome was more effective on the regulation of CD4 positive T cells than CD8 positive T cells. As for B cells, we used an antibody cocktail (F(ab)2 anti-IgM, IL-2, and anti-CD40 antibody) to stimulate original human B cells [56,57]. Then the cells were incubated with the conditional medium of each group for 7 days. The cell proliferation and apoptosis were subsequently studied. As shown in Fig. 4l and Fig. S12, for cell proliferation, compared with activated B cells, the cells in the conditional medium from the control and control + IFN- γ groups were slightly decreased, and the 60 min group was lower than the control and control + IFN- γ groups. The 60 min + IFN- γ group had the lowest number. These results indicate that the enhanced MSC secretome can effectively inhibit activated B cell proliferation. However, for cell apoptosis, all groups had no clear difference. Last, we used an ELISA assay to measure the cytokine secretion capacity of B cells from different groups. As shown in Fig. 4 m and n, the secretion of TNF- α had no significant differences among all groups. However, for IL-10, the 60 min + IFN- γ group had the highest concentration. All groups had higher concentrations than the activated B cells without conditional medium incubation ($p < 0.0001$). In conclusion, although some inconsistencies are present in the results - for example, some of the pro-inflammatory cytokines from MSCs still have comparable secretion level among all groups and the percentage of CD8 positive T cells and B cell apoptosis had no clear tendency - it is still reasonable to conclude that the MSC secretome enhanced via acoustofluidic assembly shows a more substantial immunomodulation effect than traditional MSC cultures.”

Comment 2:

I would suggest to cite in the Introduction more recently published references on the subject of MSC and their secretome application i.e PMID 24268069, 27356536, 26201487.

Response 2:

Thanks so much for your suggestion. We have added more references to better describe MSCs and their secretome applications.

Changes to the manuscript:

References added:

10. Konala, V.B.R., et al. *The current landscape of the mesenchymal stromal cell secretome: A new paradigm for cell-free regeneration. Cytotherapy* 18, 13-24 (2016).
14. Harman, R.M., Marx, C. & Van de Walle, G.R. *Translational Animal Models Provide Insight Into Mesenchymal Stromal Cell (MSC) Secretome Therapy. Frontiers in Cell and Developmental Biology* 9(2021).
15. Ferreira, J.R., et al. *Mesenchymal Stromal Cell Secretome: Influencing Therapeutic Potential by Cellular Pre-conditioning. Frontiers in Immunology* 9(2018).

Reviewer 2

Dear Author, Dear Editor, Thank you for the opportunity to share your article entitled "Acoustofluidic Interfaces for the Mechanobiological Secretome of MSCs: AIMS". In the mentioned study, you present an acoustofluidic strategy to enhance the secretion of trophic factors of MSCs. You show that the developed AIMS platform promotes the formation of 3D aggregates and detected higher concentrations of various growth factors and cytokines in the conditioned medium of 3D aggregates than in the control group (MSC cultures without acoustofluidic stimulation). You attribute the increased cytokine levels in the supernatant of the 3D aggregates to enhanced cell-cell interactions mediated by N-cadherins. Furthermore, you provide evidence that the increased levels of molecular factors in the supernatant of the 3D aggregates have immunomodulatory effects on a murine macrophage cell line (RAW 264.7 cells), which in turn reduces their pro-inflammatory secretion profile (after LPS stimulation). This effect could be enhanced by pre-stimulation of MSCs with IFN- γ . In summary, this study confirms the well-established concepts that the secretory properties of MSCs are significantly affected by the amount of cell-cell contact and that stimulation of cells with IFN- γ increases their immunomodulatory phenotype. For the most part, the quality of the data is technically sound and presented in sufficient detail; however, the novelty value of the results is limited and advances our understanding of the mechanisms underlying the secretory (or immunomodulatory) properties of MSCs only to a limited extent. Also, in terms of potential therapeutic application of the results, fear to say that your study provides little evidence to advance the field. The use of the AIMS platform for basic research questions is very promising, but mainly of interest to a specialized audience.

Response:

We thank the reviewer for the constructive feedback. We have addressed each of your comments, added more experiments to better describe our AIMS platform for MSC secretome improvement, and revised the manuscript accordingly.

Comment 1:

The formation of MSC aggregates with the AIMS platform is very interesting, but it is important to introduce further controls to classify the results. MSCs also form 3D spheroids in ultra-low attachment plates, which would allow discrimination between the influence of 3D structure and acoustofluidic stimulation. An alternative control (which is mandatory) is MSCs cultured on polyvinyl alcohol treated plates but not acoustofluidically stimulated.

Response 1:

The reviewer raises a very good point. Indeed, a 3D-culture group is a necessary control to compare normal 3D spheroids and acoustofluidic-treated cell aggregates. According to the reviewer's suggestion, we used commercialized Spheroid Microplates (Corning®, 10185) to 3D-culture MSCs as another reference group. We repeated all the experiments, and we found that although the MSC viability of the 3D Microplate group after 4 days of incubation was lower than other groups, after normalizing by DNA content, the relative secretome profile of this group was still comparable with the 10 min acoustofluidic-treated group. Therefore, the 3D culture of MSCs in the microplates indeed also improved MSC secretome. However, the 30 and 60 min acoustofluidic-treated groups still had an overall better secretome profiles than other groups, which indicated that a long-term duration of acoustofluidic treatment was more helpful for MSC secretome. This may be attributed to better cell-cell interaction. According to our flow cytometry results, 30 and 60 min acoustofluidic assembly groups had higher expressions of N-cadherin markers. Therefore, we consider that our AIMS is a better platform for MSC secretome improvement than commercialized Spheroid Microplates.

Comment 2:

The differences in cytokine/growth factor concentrations in the conditioned supernatant of the different cultures are very interesting. However, to what extent can you rule out that the observed differences are not due to differences in cell number? You write, "The cells in all groups showed a good proliferation profile, and no short-term adverse effects were observed in the 30- and 60-minute groups. The assay used for this purpose (CCK-8) is suitable for adherent monolayer cells, but not for 3D spheroids. Studies by other colleagues have already shown that the assay is inappropriate for the determination of cell number in 3D spheroids. More direct methods such as DNA content determination (e.g., using CyQuant proliferation assays) or direct cell counting after dissociation of the 3D structure are mandatory to determine the exact cell number (see <https://doi.org/10.1016/j.jtcms.20a20.09.004> - especially Figures 3 and 4). Figures 3b (FDA staining) and 3c (Hoechst) clearly show the differences in cell number between cultures within the first 2 days.

Response 2:

Thanks for the reviewer's helpful suggestion. Per the reviewer's comments, we replaced the former CCK-8 assays with CyQuant proliferation assays to determine the total DNA contents of each group. To avoid the influence from cell number difference, we used DNA contents to normalize the secretome profile. In Figs. 2-4, we performed DNA content measurements before each experiment. According to our results, shown in Fig. 1 below, only the 3D Microplate group showed lower cell viability than other groups after 4 days of incubation, which was also supported by another study (*Modulation of Inherent Niches in 3D Multicellular MSC Spheroids Reconfigures Metabolism and Enhances Therapeutic Potential. Cells 10, 2747 (2021)*). The study claimed that the formation of MSC spheroid would limit the oxygen diffusion, and the cell viability was affected even after 3 days. For the FDA staining images in Fig. 3, acoustofluidic assembly concentrated the cells at the center of the field, and cells in the control group spread over the substrate. After acoustofluidic assembly, the cells also formed a 3D layer-by-layer structure, which may indicate some visual errors. According to our DNA content measurements, statistically cell viability in our AIMS platform was not affected.

Figure 1 | e, CyQUANT™ Cell Proliferation Assay measuring the DNA contents of different groups. Data are graphed as the mean \pm SD (n=4), *p < 0.05, **p < 0.01.

Changes to the manuscript (Page 5 in the main text):

Cells cultured in the commercialized Spheroid Microplates were denoted as 3D Microplates. We used a CyQUANT™ Cell Proliferation Assay to directly measure the total DNA content of each group. As shown in Fig. 1e, after 1 day of incubation, there was no significant difference of DNA contents among all groups. However, after 4 days of incubation, the DNA content of the 3D Microplate group was lower than other groups, meaning that the commercialized Spheroid Microplates was not suitable for long-term MSC proliferation" in Results section, and "We also compared our AIMS platform with commercialized 3D cell culture microplates to distinguish the potential impacts from cell 3D culture and acoustofluidic stimulation. We found MSC proliferation in 3D microplate was inhibited, which was also supported by another study, which claimed that the formation of MSC spheroid would limit the oxygen diffusion, and the cell viability was affected even after 3 days [66]. Cell viability in our AIMS platform was not affected statistically, which may be attributed to the similar culture condition of AIMS with conventional cell culture substrates.

Comment 3:

The secretion profiles and exosome diameter profiles are very different between the acoustofluidically stimulated groups. Is there an explanation for this? Does this have a functional impact (e.g., on immunomodulatory effects?).

Response 3:

Indeed, the secretion profiles and exosome diameter profiles showed differences. In the first version of the manuscript, the cytokine array assay was only performed for one time and presented as a heatmap image. In addition, the expression period of different cytokines may be different. For the NTA analysis of exosome diameters, we also only used one sample for each group. Per the reviewer' comments, we prepared four repeats for each group and collected them together. Then we used the collected medium to perform the cytokine array assay. We also extended the incubation time to four days. For NTA analysis, we repeated each sample five times and generated the diameter profiles with error bars, as shown below:

Figure 2b, Heatmap presenting the secretome profiles of different groups after incubating for 4 days. The data was processed by the Z-score normalization and clustering analysis.

Figure 2o, NTA test measuring the diameters of secreted exosomes from different groups.

According to our results and consistent with the first version of the manuscript, for the secretome profile, the 60 min group showed the best outcome. The 30 min group also had an overall higher expression of the cytokines than the control, 3D Microplate, and 10 min groups. There were still a few abnormal targets (such as IL-3 and IL-12) that showed inconsistent tendency. There are several potential reasons for this: first, the heatmap results come from the single integrated densities of each target, although we collected four repeats together for each group. Second, the accuracy of cytokine array assay is not as good as an ELISA assay. Therefore, using an ELISA assay to detect more cytokines will largely support the results. For the exosome diameter analysis, we added error bars in the figure. The diameters of each group were still within the 50-150 nm range.

Changes to the manuscript (Page 6 in the main text):

The concentrations of a broad spectrum of soluble proteins - comprised of 80 total targets (78 available), such as growth factors and chemokines - were investigated using a cytokine assay to analyze the cytokine secretion profiles (Fig. 2b). For the 60 min group, 57 targets had the highest concentration compared to other groups, and 14 targets had the second-highest

concentration. For the 30 min group, 14 targets had the highest concentration, and 44 targets had the second-highest concentration. However, for the 10 min group, only 6 targets had the highest concentration and 10 targets had the second-highest concentration. As for the 3D Microplate group, after normalized to the same DNA content, only 1 target had the highest concentration and 9 targets had the second-highest concentration. In addition, 65 targets had the lowest concentration in the control group. In general, the 3D Microplate, 10 min, 30 min, and 60 min groups improved the secretome profile of MSCs, compared to the control group. The 60 min group had the best effect. However, it is also worth noting that the original integrated density of these four groups only had slight differences (Fig.S1), which indicated that their secretome profiles were relatively similar." and "The exosome characterizations, including morphology and diameter measurement, are shown in Fig. 2n and o. The exosomes collected following acoustofluidic assembly showed a typical globular shape, and their diameters were within the 50-150 nm range.

Comment 4:

There is a lack of critical discussion of the secretion data - known proinflammatory factors (TNF- α , IFN- γ , IL-6) have much higher levels in the AIMS cultures than in the controls - how can this be reconciled with immunomodulatory effects?

Response 4:

According to our results, the secretome profile after acoustofluidic assembly had an overall improvement, including growth factors, pro-inflammatory factors, anti-inflammatory factors, *etc.* Therefore, it is important to pretreat MSCs before using them for immunomodulation applications. Using IFN- γ to prime MSCs and help them enter the immunomodulatory state is a feasible way for the therapeutic practice of MSCs. After priming by IFN- γ , we found the secretion of anti-inflammatory factors was further improved. The primed MSCs in our AIMS platform had the highest concentrations of anti-inflammatory factors. Although there were some inconsistencies for the secretion of pro-inflammatory factors and chemokines, it was still efficient to inhibit the pro-inflammatory response of M1 phenotype macrophages, suppress T cell activation, and support B cell functions. Per the reviewer's comment, we have added this discussion in the manuscript.

Changes to the manuscript (Page 16 in the main text):

According to our results, the secretome profile after acoustofluidic assembly had an overall improvement, including growth factors, pro-inflammatory factors, anti-inflammatory factors, etc. Other studies also reported that the paracrine effect of MSCs was enhanced in a full scale. Therefore, using IFN- γ to pretreat MSCs and help them enter the immunomodulatory state was essential for the therapeutic practice of MSCs. After priming by IFN- γ , we found the secretion of anti-inflammatory factors was further improved. The primed MSCs in our AIMS platform had the highest concentrations of anti-inflammatory factors. Although there were some inconsistencies for the secretion of pro-inflammatory factors and chemokines, the c was still efficient to inhibit the pro-inflammatory response of M1 phenotype macrophages, suppress T cell activation, and support B cell functions.

Comment 5:

The qualitative differences in the amount of N-cadherin are remarkable. Could you please also present the quantitative differences at mRNA and protein level normalized to the one

housekeeping gene/protein or cell number to make the interpretation of the results comprehensible.

Response 5:

We performed Western blotting, flow cytometry, and QPCR to characterize the expression of N-cadherin as below:

Figure 3 l, Western blotting detecting the expression of N-cadherin of MSCs from different groups. m, Flow cytometry analysis measuring the expression of N-cadherin of MSCs from different groups.

Figure S5 | QPCR measuring the relative expression of CDH2 gene (N-cadherin) of different groups after incubating for 3 days. Data are graphed as the mean ± SD (n=3, biological repeats), ***p < 0.001.

According to our results, the 3D Microplate, 10 min, 30 min, and 60 min groups had a clear increase of N-cadherin expression compared to the control group. For the flow cytometry results, 60 min group had the highest N-cadherin expression level. 3D Microplate, 10 min, and 30 min groups also had higher levels than the control group. For the QPCR result, although all the 3D Microplate, 10 min, 30 min, and 60 min groups still had more CDH2 gene expressions than the control group, and these four groups had no significant difference. Even though there was some

inconsistency between the genetic and protein level, it is still reasonable to conclude that the formation of the MSC spheroids up-regulates the expression of N-cadherin, improving the secretome profile of the MSCs.

Changes to the manuscript (Page 10 in the main text):

In general, 3D Microplate, 10 min, 30 min, and 60 min groups had a clear increase of N-cadherin expression compared to the control group. According to the flow cytometry results, 60 min group had the highest N-cadherin expression level. 3D Microplate, 10 min, and 30 min groups also had higher levels than the control group. We also use QPCR assay to detect N-cadherin expression at the genetic level. As shown in Fig. S5, although all the 3D Microplate, 10 min, 30 min, and 60 min groups still had more CDH2 (N-cadherin) gene expressions than the control group, but these four groups had no significant difference. Even though there was some inconsistency between the genetic and protein level, it is still reasonable to conclude that the formation of the MSC spheroids up-regulates the expression of N-cadherin, improving the secretome profile of the MSCs.

Comment 6:

After functional blocking of N-cadherin, the secretion profiles of the different groups change but are not identical. Is it correct that the control was also treated with N-cadherin antibodies? Could you please add corresponding reference samples (e.g. 60 min and control without N-cadherin blocking) to the heatmap?

Response 6:

We thank the reviewer for suggesting this addition to the manuscript. Per the reviewer's comments, we have repeated the cytokine array assay with the addition of 60 min and control groups:

Figure 3b, Heatmap measuring the secretome profiles of different groups after functional blocking of N-cadherin of MSCs and incubating for 4 days. The data was processed by the Z-score normalization and clustering analysis.

According to our results, the control and 60 min groups without neutralizing antibody treatment present a similar tendency to the results in Fig. 2. After functional blocking of N-cadherin, the tendency of all groups was not clear, in agreement with the former manuscript. Indeed, for each group, the secretome profiles are not identical. Because the heatmaps in Fig. 2 and Fig. 3 were processed by the Z-score normalization, we presented the original data of integrated densities in Fig. S1 and Fig. S3, as shown below:

Figure S1 | Heatmap presenting the integrated densities of different groups after incubating with the cytokine array for 4 days (without Z-score normalization and clustering process).

Figure S3 | Heatmap presenting the integrated densities of different groups after functional blocking N-cadherin of MSCs and incubating with the cytokine array for 4 days (without Z-score normalization and clustering process).

We find that the original integrated densities of each group for these targets were pretty close. As we mentioned before, we prepared four repeats for each group and collected them together. Then we used the collected medium to perform the cytokine array assay. The expression period of different proteins may be different, and the accuracy of the cytokine array assay is not as good as an ELISA assay. Therefore, it is good for understanding the overall tendency by the cytokine array assay. It should be better to perform an ELISA assay to accurately and quantitatively distinguish the actual expression differences among different groups.

Changes to the manuscript (Page 9 in the main text):

We used an N-cadherin neutralizing antibody (nAb) to observe if functionally blocking N-cadherin affected the secretome profile. After pre-treated by nAb, we performed 3D Microplate culture and acoustofluidic assembly for the cells. We also used normal MSCs without nAb treatment as the control and 60 min groups. Same to above, the total DNA content of each group was measured and used for data normalization (Fig. S2). Firstly, we conducted the cytokine array assay again (79 targets available), with the results shown in Fig. 3b. Similar to previous results, the normal control group and 60 min group showed a clear difference. However, for the nAb pre-treated

groups, it is evident that the cytokine secretion profiles of each group showed no significant changes, unlike the results obtained in Fig. 2b that contained active N-cadherin, suggesting that the acoustofluidic-induced enhancements to the MSC secretome were eliminated after functionally blocking N-cadherin. The original data (heatmap without z-score normalization) showed that the integrated densities of each group were very close (Fig. S3), although some of the targets in 30 min + nAb and 60 min + nAb were slightly higher than other nAb pre-treated groups. Next, we repeated the ELISA assay for the 8 targets to examine whether the improved secretome was abolished. As shown in Fig.2c-j, same to cytokine array assay, the cytokine secretion had a similar tendency among all groups.

Comment 7:

What is the impact of functional blocking of N-cadherin on the immunomodulatory properties of the conditioned medium? Does it also affect IFN- γ licensing of the MSCs?

Response 7:

To better explain this question, we performed the ELISA assay for 8 targets including FGF-2, IGF-1, VEGF, HGF-1, IL-6, IL-10, IFN- γ , and TNF- α . As shown below:

Figure 3 | c-j, ELISA assay measuring the secretion of FGF-2, IGF-1, VEGF, HGF-1, IL-6, IL-10, IFN- γ , and TNF- α from different groups after functional blocking of N-cadherin of MSCs and incubating for 4 days. Data are graphed as the mean \pm SD (n=4), * p < 0.05, ** p < 0.01, *** p < 0.001, **** p < 0.0001.

According to our results, after blocking of N-cadherin, the expression of these targets from each group had no significant difference (compared with the results in Fig. 4, in which after primed by IFN- γ , the secretion of growth factors and anti-inflammatory factors was significantly improved). Since these cytokines play very important roles in MSC immunomodulatory functions, we thought the immunomodulatory function is not as good as the medium from the cells after being primed by IFN- γ . The cells we used for IFN- γ pre-treatment were original MSCs without N-cadherin blocking. Therefore, the N-cadherin blocking would not influence the IFN- γ licensing and the results in Fig. 4.

Changes to the manuscript (Page 9 in the main text):

Next, we repeated the ELISA assay for the 8 targets to examine whether the improved secretome was abolished. As shown in Fig.3c-j, similar to the cytokine array assay, the cytokine secretion had a similar tendency among all groups.

Comment 8:

In the studies on immunomodulatory effects, control monolayer cultures with IFN- γ stimulation are missing. Without this control, evaluation and interpretation regarding the influence of AIMS are difficult. In other words, IFN- γ stimulation could in any case be sufficient to produce a comparable effect in monolayers without AIMS.

Response 8:

Thanks for the reviewer's helpful suggestion. Per the reviewer's comment, we have repeated the immunomodulation-related experiments with the control + IFN- γ group, as shown below:

Figure 4 | Harnessing the enhanced MSC secretome for immunomodulation applications. **a**, Heatmap measuring the inflammatory factors secreted from IFN- γ primed MSCs from different groups. **b-i**, ELISA assay measuring the secretion of FGF-2, IGF-1, VEGF, HGF-1, IL-6, IL-10, IFN- γ , and TNF- α from different groups after incubating for 4 days. **j**, Flow cytometry analysis measuring the percentage of CD86 and CD163 positive human THP-1 cells after incubating with the conditional medium from MSCs for 2 days. **k**, Flow cytometry analysis measuring the percentage of CD4 positive T cells after incubating with the conditional medium from MSCs for 3 days. **l**, Flow cytometry analysis measuring the B cell proliferation after incubating with the conditional medium from MSCs for 7 days. **m&n**, ELISA assay measuring the secretion of TNF- α and IL-10 from B cells. Data are graphed as the mean \pm SD (n=4), *p < 0.05, **p < 0.01, ***p < 0.001, ****p < 0.0001.

According to our results, MSCs in the control + IFN- γ group can secrete more anti-inflammatory factors and growth factors than the control group. The secretion of pro-inflammatory factors and chemokines are complicated. Some targets had higher concentrations in the control + IFN- γ group, but some were lower, compared with the control group. We used the conditional medium to study the influence on phenotype switch of macrophages, T cell activation, and B cell functions. We found that for all these experiments, the control + IFN- γ group had a slightly better

effect than the control group, but they were not good as the 60 min and 60 min + IFN- γ group, which may be attributed to the lower cytokine concentrations (seen in ELISA results).

Changes to the manuscript (Page 11 in the main text):

In Fig. 4a, pro-inflammatory cytokines are presented as red color and anti-inflammatory cytokines are presented as blue color. The tendency of anti-inflammatory cytokines was very clear; the 60 min + IFN- γ group had the highest concentration. The control + IFN- γ group also had higher concentrations than the control group, which indicated that the IFN- γ pretreatment can stimulate MSCs to secrete more anti-inflammatory cytokines. The results for pro-inflammatory cytokines and chemokines are not very consistent. There were 16 targets that the IFN- γ pretreatment would decrease their secretion, including EOTAXIN, EOTAXIN-2, IFN- γ , IL-1 β , IL-2, IL-3, IL-6sR, IL-7, IL-12p70, IL-17, IP-10, MIGMI, MIG, MIP-1 β , MIP-1 δ , and TNF- α . For these targets, the 60 min group had the highest concentration, and the control group had the second-highest concentration. After the IFN- γ pretreatment, these targets in 60 min + IFN- γ group were lower than the control group. The control + IFN- γ group had the lowest concentration. There were also 9 targets that the IFN- γ pretreatment did not change their secretion profile, including GCSF, GM-CSF, ICAM-1, I-309, IL-1 α , IL-12p40, IL-15, MCP-1, and MCP-2. For these targets, both 60 min and 60 min + IFN- γ groups had higher concentrations than the control and control + IFN- γ groups. We also repeated the above-mentioned ELISA assay for the 8 targets to better observe the cytokine secretion profile. As shown in Fig.4b-i, FGF-2, IGF-1, VEGF, HGF-1, and IL-10 showed a similar tendency. The 60 min + IFN- γ group had the highest concentration. It was much higher than the control and control + IFN- γ groups ($p < 0.001$). The 60 min group was also higher than the control and control + IFN- γ groups ($p < 0.01$). For IL-6, IFN- γ , and TNF- α , the 60 min group had the highest concentration. The other three groups had similar profiles. According to our results, consistent with the results in Fig.2, the 60 min group can improve the secretion of both pro- and anti-inflammatory factors. For anti-inflammatory factors and growth factors, the IFN- γ pretreatment of MSCs can further improve their secretion. However, for pro-inflammatory factors and chemokines, some of the targets were not influenced by the IFN- γ pretreatment.

Comment 9:

The presented heatmap (figure 4b) seems to contradict in essential parts the heatmap in figure 2b? The analytes IL-13, TNF- α , TNF- β , TGF- β , RANTES, MCP-1, IP-10, IL-16, IL-12 p40/p70, IL-3, IFN- γ are significantly higher in the 60 min group than in the control in figure 2b, while it is exactly the opposite in figure 4b? IL-10 is comparable between 60min group and control in Figure 2b and not different as in Figure 4b. Finally, MIP-1 β in Figure 2b is higher in the control than in the 60 min group, while in Figure 4b it is the other way around. In addition to the heatmaps in Figure 2, 3 and 4, please show quantitative (ELISA / multiplex ELISA) data for selected (but the same) cytokines as surrogate markers for the secretome. I would suggest based on your reasoning: TNF- α , IFN- γ , IL-6, IL-10, FGF-2, VEGF, and HGF. This would make the figures comparable to each other and better justify your argumentation.

Response 9:

Thanks for the reviewer's helpful comment. We also noticed these inconsistencies between different measurements. As we mentioned before, in the first version of the manuscript, the cytokine array assay was only performed one time (presented as a heatmap image). In addition, the expression period of different cytokines may be different. To improve our results, we prepared four repeats for each group and collected them together. Then we used the collected medium to

perform the cytokine array assay. We also extended the incubation time to four days. We have replaced the new heatmap figures. However, there were still a few inconsistencies or abnormal targets. We also did the ELISA assay for eight targets (As shown in Fig. 4 in Comment 8). According to our results, the secretion profile of control and 60 min groups was similar to the results in Fig. 2. For the control + IFN- γ and 60 min + IFN- γ groups, as we mentioned before, the secretion of anti-inflammatory factors and growth factors of 60 min + IFN- γ group was higher than 60 min group. The control + IFN- γ group was higher than the control group. For pro-inflammatory and chemokines, the 60 min group had the highest concentration. The 60 min + IFN- γ group was similar to the control group.

Changes to the manuscript (Page 11 in the main text):

In Fig.4a, pro-inflammatory cytokines were presented as red color, anti-inflammatory cytokines were presented as blue color. The tendency of anti-inflammatory cytokines was very clear, the 60 min + IFN- γ group had the highest concentration. The control + IFN- γ group also had higher concentrations than the control group, which indicated that the IFN- γ pretreatment can stimulate MSCs to secrete more anti-inflammatory cytokines. The results for pro-inflammatory cytokines and chemokines are not very consistent. There were 16 targets that the IFN- γ pretreatment would decrease their secretion, including EOTAXIN, EOTAXIN-2, IFN- γ , IL-1 β , IL-2, IL-3, IL-6sR, IL-7, IL-12p70, IL-17, IP-10, MIGMI, MIG, MIP-1 β , MIP-1 δ , and TNF- α . For these targets, the 60 min group had the highest concentration, and the control group had the second-highest concentration. After the IFN- γ pretreatment, these targets in 60 min + IFN- γ group were lower than the control group. And the control + IFN- γ group had the lowest concentration. There were also 9 targets that the IFN- γ pretreatment did not change their secretion profile, including GCSF, GM-CSF, ICAM-1, I-309, IL-1 α , IL-12p40, IL-15, MCP-1, and MCP-2. For these targets, both 60 min and 60 min + IFN- γ groups had higher concentrations than the control and control + IFN- γ groups. We also repeated the above-mentioned ELISA assay for the 8 targets to better observe the cytokine secretion profile. As shown in Fig.4b-i, FGF-2, IGF-1, VEGF, HGF-1, and IL-10 showed a similar tendency. The 60 min + IFN- γ group had the highest concentration. It was much higher than the control and control + IFN- γ groups ($p < 0.001$). The 60 min group was also higher than the control and control + IFN- γ groups ($p < 0.01$). For IL-6, IFN- γ , and TNF- α , the 60 min group had the highest concentration. Other three groups had the similar profiles. According to our results, consistent with the results in Fig.2, the 60 min group can improve the secretion of both pro- and anti-inflammatory factors. For anti-inflammatory factors and growth factors, the IFN- γ pretreatment of MSCs can further improve their secretion. However, for pro-inflammatory factors and chemokines, some of the targets were not influenced by the IFN- γ pretreatment.

Comment 10:

Regarding 4a: Why should anti-inflammatory therapy be used to treat an infection?

Response 10:

We corrected the inaccurate description. In the current version, the term "immunomodulation applications" was used.

Changes to the manuscript (Page 11 in the main text):

Harnessing the enhanced MSC secretome for immunomodulation applications

Comment 11:

Why did you use a murine cell line to study the effects of conditioned medium from human cells?

Response 11:

We are thankful for reviewer's kind reminding. We have used human THP-1 cells to repeat all the experiments. For human THP-1 cells, cells were cultured in RPMI 1640 medium (Gibco, 72400120) with 10% of FBS (v/v). Phorbol-12-myristate 13-acetate (PMA) (Sigma-Aldrich, 100 nM) was used to differentiate THP-1 cells to macrophages. Then the cells were treated by 100 ng/mL of lipopolysaccharide (LPS, Sigma-Aldrich, L2018), dissolved in culture medium for 12 h, to induce the M1 polarization. For the following immunomodulation experiments, the conditioned medium was mixed with RPMI 1640 medium (1: 1, v/v) to culture the LPS-treated THP-1 cells. The updated Fig. 4 was shown before. According to our results, for CD86, the positive percentage of the LPS pretreatment, control, control + IFN- γ , 60 min and 60 min + IFN- γ groups were 48.2, 45.4, 40.2, 39.5, and 27.3%, respectively. Although the effects were not that remarkable compared with the effects used urine RAW264.7 cells, the human THP-1 cells still presented similar tendency to RAW264.7 cells.

Comment 12:

The qualitative assessment of macrophage polarization by CD86 is good, however, 2b macrophages also express CD86 - does the conditioned medium of MSCs change the macrophage polarization from M1 to M0 or to M2. Please try to generate quantitative data and use a larger panel (e.g. MHCII, CD68, CD80, CD86, CD163) to allow classification of results.

Response 12:

We have used CD80, CD86, and CD163 markers to perform the flow cytometry assay. CD80 was considered as a general macrophage marker when THP-1 cells were pre-treated by PMA. Then we used both CD86 and CD163 to distinguish CD80+ & CD86+ and CD80+ & CD163+ cells. The results are shown below:

Figure 4 | j, Flow cytometry analysis measuring the percentage of CD86 and CD163 positive human THP-1 cells after incubating with the conditional medium from MSCs for 2 days.

Changes to the manuscript (Page 13 in the main text):

Next, flow cytometry assay was used to detect the phenotype switch of THP-1 cells (Fig. 4j). After being induced by PMA, all the cells expressed macrophage marker CD80. We used CD86 and CD163 as the M1 and M2 phenotype markers. For CD86, the positive percentage of the LPS pretreatment, control, control + IFN- γ , 60 min and 60 min + IFN- γ groups were 48.2, 45.4, 40.2, 39.5, and 27.3%, respectively. The number of CD86 positive cells gradually decreased. As for CD163, the positive percentage were 9.92, 12.1, 12.9, 14.4, and 19.1%, respectively. The number of CD163 positive cells increased. In other words, the 60 min + IFN- γ group had the largest number of M2 phenotype macrophages and lowest number of M1 phenotype macrophages. The control + IFN- γ group also had larger amount of M2 phenotype macrophages and less M1 phenotype macrophages compared to the control group, which meant that the IFN- γ pretreatment indeed was effective in inducing the immunomodulative activities of MSCs.

Comment 13:

Please indicate if the N numbers in the captions are biological or technical replicates. MSCs from how many donors were used? - could you provide age and sex information?

Response 13:

The N number in this manuscript was biological repeats. We have added the information in every figure caption. We have added the donor information in the Supplementary Information.

Comment 14:

The statistical analysis is not sufficient. Was the Student's t-test two-sided? Student's t-test is only permissive for comparisons of two groups. For comparisons of more than two (independent) groups, it is mandatory to adjust the alpha levels and apply an appropriate statistical test with appropriate correction procedures for the p-value for multiple comparisons.

Response 14:

We apologize for the mistake. We have corrected the statement of statistical analysis. The method we used was one-way ANOVA with Tukey's post-hoc test to compare multiple columns in each figure.

Changes to the manuscript (Page 22 in the main text):

*The statistical analysis was performed using one-way ANOVA with Tukey's post-hoc test by GraphPad Prism 9.0. The data is presented as a mean \pm standard deviation (SD). Confidence levels of * $p < 0.05$, ** $p < 0.01$, *** $p < 0.001$, and **** $p < 0.0001$ were selected as the threshold values.*

Comment 15:

Please try to have a more balanced discussion in relation to biomaterials. For clinical application of cell therapy, biomaterials are often necessary to allow transplantation of cells in the first place and to keep them at the site of injury (carrier function), furthermore biomaterials protect

regenerative cells from negative influences of local immune cells (support function). Finally, many MSC isolations express tissue factor, which precludes systemic administration of the cells, as this can be a high risk for blood-mediated inflammatory reactions (IBMIR).

Response 15:

Biomaterial research helped us significantly in this manuscript. Per the reviewer's comment, we have added the related sentences in this Discussion section.

Changes to the manuscript (Page 14 in the main text):

Undeniably, biomaterials and tissue engineering play an irreplaceable role in stem cell therapy, especially in vivo implantations. For example, material scaffolds can provide stable support and a gentle environment for MSCs to avoid the side effects of immune reactions or interferences from other biological components. MSCs can thus exert their functions effectively over the long term.

Comment 16:

Please also discuss the potential application of your results in terms of clinical and basic research. What would be the therapeutic advantage of using conditioned medium instead of cell therapy? Proteins have a much shorter half-life than living cells? Biomaterials have the advantage that they can stimulate or inhibit specific mechanisms of cells in 3D.

Response 16:

Per the reviewer's comment, we have added the related sentences in this Discussion section.

Changes to the manuscript (Page 17 in the main text):

The improved secretome could be a good alternative to stem cell therapy, as potential risks associated with stem cell transplantation, such as immune rejection, tumorigenicity, or unwanted differentiation, are significantly reduced or eliminated. The production of conditioned medium can be easily scaled up, ensuring a consistent and standardized treatment option, which may largely avoid the shortage of stem cell supply in clinic. The improved secretome can also be integrated with biomaterials for implantation to support in situ tissue regeneration consistently and prevent the early degradation of proteins.

Reviewer 3

This paper applies a method based on acoustic trapping/streaming to generating 3D cultures of MSCs and harvesting the excreted proteins and exosomes (secretome). The novel aspect is that the effect of acoustofluidic culture on the composition of the secretome has been investigated. The authors suggest that this composition may be influenced by culture in the acoustic trap. The paper is timely, proposes an interesting device and concept, and provides a very interesting characterisation of the excreted proteins and exosomes. In the paper the phrase improved secretome is used often but not well defined. Supposedly the authors mean a secretome with a composition that would be more effective if used therapeutically? If this is the claim, a clearer definition of what this means in terms of specific molecules, or for instance over-all concentration levels, needs to be supplied or discussed in more detail. Alternatively, a more nuanced way of talking about the changes to the secretome may be sufficient. Key aspect of assessing the originality is to see what effects on the secretome that can be attributed to the acoustic trapping/streaming (which may add for instance mechanical stimulation and mixing) and what is the general effect of 3D culture. For instance,

reference 27 seems to suggest that large effects on the secretome can be found in a generic 3D culture. Would there be a significant difference when comparing the method to for instance 3D culture in a well plate with conical bottom, a hanging droplet culture or similar? All of the controls in the paper are made to a monolayer culture so it is hard to say whether the improvements are general 3D culture features or specific to the acoustic trap. If the enhancement of the secretome could be accomplished on any 3D culture platform those may be significantly easier to scale for parallel operation which would be needed if this was to be used therapeutically. On the other hand, if the effects on the secretome was specific to the acoustic setup that would be highly interesting. To evaluate this, I think additional experiments may be needed.

Response:

We sincerely thank the reviewer for the helpful suggestions. We have used a commercialized 3D-culture microplate to compare with our AIMS platform. We have addressed each of your comments and revised the manuscript accordingly.

Comment 1:

A comparison with conventional spheroid culture may make it possible to assess that the effects of the AIMS method beyond what is accomplished in any system providing 3D aggregation.

Response 1:

We are thankful for the reviewer's helpful suggestion. We have used commercialized Spheroid Microplates (Corning®, 10185) to 3D-culture MSCs as another control group. We repeated all the experiments, and we found although the MSC viability of the 3D Microplate group after 4 days of incubation was lower than other groups, however, after being normalized by DNA content, the relative secretome profile of this group was still comparable with the 10 min acoustofluidic assembly group. Therefore, the 3D-culture of MSCs in the microplates indeed also improved MSC secretome. However, the 30 and 60 min acoustofluidic assembly groups still had overall better secretome profiles than other groups, which indicated that a long-term duration of acoustofluidic treatment was more helpful for MSC secretome, which may be attributed to a better cell-cell interaction. According to our flow cytometry results, the 30 and 60 min acoustofluidic assembly groups had higher expressions of N-cadherin markers. Therefore, we consider that our AIMS is a better platform for MSC secretome improvement.

Comment 2:

It would be interesting to further explore a wider range of actuation times. The paper describes how the cells are aggregated within 90 s. A comparison between turning the actuation off directly after aggregation to having the aggregation on for an entire assay (days) might also provide interesting results for analysing the effect of the acoustic forces.

Response 2:

We appreciate the reviewer's helpful comment. Actually, when we did the preliminary test, we found that 10 min of acoustofluidic treatment was the minimum time for the formation of cell aggregates. If we turn off the acoustics once after the aggregation, the aggregate morphology cannot be well maintained during the incubation. There were several reasons that it was difficult to keep acoustics on for several days: first, the droplet volume we used was 200 μ L (due to the space limitation of our device, which need to match the size of the transducer) and the cell viability and normal growth would be severely affected by long-term acoustic treatment. Second, long-term acoustic treatment would be a huge burden for not only the experiments we did (four

repeats for each group, three-five groups for different experiments), but it would also not be very feasible for clinical use. Therefore, in this manuscript, we decided to distinguish the potential effects of acoustofluidic treatments by 10, 30, and 60 min. Indeed, we found that these different time durations had distinct outcomes. In the future, we could do more experiments to customize a large-scale setup and optimize the whole process to continue explore the long-term influence on stem cell behaviors (including osteogenic differentiation, apoptosis, cell cycle, etc)

Comment 3:

The acoustofluidic device presented is very interesting in itself, with a minimalistic and straight-forward configuration. It is not clear however if the mechanism for particle manipulation is based on acoustic streaming or radiation forces. In the paper the authors seem to suggest that acoustic streaming is the primary mechanism, however, in a previous publication: (1) Oberti, S.; Neild, A.; Dual, J. Manipulation of Micrometer Sized Particles within a Micromachined Fluidic Device to Form Two-Dimensional Patterns Using Ultrasound. *J. Acoust. Soc. Am.* 2007, 121 (2), 778–785. <https://doi.org/10.1121/1.2404920>. A similar configuration where surface modes on a plate are also used to manipulate particles radiation forces are more important. It would be interesting if the authors could elaborate on why streaming is the most important mechanism or whether it is a combination effect. I found very little discussions and citations to the prior literature concerning acoustic trapping and formation of 3D cultures. Very much has been done on this topic and in addition acoustic forces has been used to form 3D cultures of MSCs specifically (Jeger-Madiot et al.). Some relevant references on this topic could include: Pioneering work: (1) Liu, J.; Kuznetsova, L. A.; Edwards, G. O.; Xu, J.; Ma, M.; Purcell, W. M.; Jackson, S. K.; Coakley, W. T. Functional Three-Dimensional HepG2 Aggregate Cultures Generated from an Ultrasound Trap: Comparison with HepG2 Spheroids. *J. Cell. Biochem.* 2007, 102 (5), 1180–1189. <https://doi.org/10.1002/jcb.21345>. A review: (2) Olofsson, K.; Hammarström, B.; Wiklund, M. Ultrasonic Based Tissue Modelling and Engineering. *Micromachines* 2018, 9 (11), 594. <https://doi.org/10.3390/mi9110594>. Recent advances/devices: (3) Jeger-Madiot, N.; Arakelian, L.; Setterblad, N.; Bruneval, P.; Hoyos, M.; Larghero, J.; Aider, J. L. Self-Organization and Culture of Mesenchymal Stem Cell Spheroids in Acoustic Levitation. *Sci. Rep.* 2021, 11 (1), 1–8. <https://doi.org/10.1038/s41598-021-87459-6>. (4) Luo, Y.; Gao, H.; Zhou, M.; Xiao, L.; Xu, T.; Zhang, X. Integrated Acoustic Chip for Culturing 3D Cell Arrays. *ACS Sensors* 2022, 7 (9), 2654–2660. <https://doi.org/10.1021/acssensors.2c01103>. (5) Chen, K.; Wu, M.; Guo, F.; Li, P.; Chan, C. Y.; Mao, Z.; Li, S.; Ren, L.; Zhang, R.; Huang, T. J. Rapid Formation of Size-Controllable Multicellular Spheroids: Via 3D Acoustic Tweezers. *Lab Chip* 2016, 16 (14), 2636–2643. <https://doi.org/10.1039/c6lc00444j>. (5) Olofsson, K.; Carannante, V.; Ohlin, M.; Frisk, T.; Kushiro, K.; Takai, M.; Lundqvist, A.; Önfelt, B.; Wiklund, M. Acoustic Formation of Multicellular Tumor Spheroids Enabling On-Chip Functional and Structural Imaging. *Lab Chip* 2018, 18 (16), 2466–2476. <https://doi.org/10.1039/C8LC00537K>. I would like to see a discussion about these methods in the paper and how the proposed design is different and perhaps advantageous for this particular application.

Response 3:

We appreciate the reviewer's helpful comment. In most instances of particle manipulation, both acoustic radiation force and drag force from acoustic streaming play a significant role. However, in the context of our method, it's the drag force that primarily facilitates cell concentration towards the device center. We provided visual aids in the form of Fig. S13 (shown below) to further clarify this. Fig. S13a illustrates the distribution of the acoustic pressure amplitude, indicating a higher value compared to surrounding areas. This creates an acoustic radiation force that radiates from

the center towards the outer regions. Conversely, Fig. S13b highlights the creation of acoustic streaming by these waves, where a low velocity region in the center is evidenced. The flow lines of acoustic streaming extend from the periphery to the center, assisting in the transportation of cells to the central area, consequently forming a rounded cell cluster. The interplay of these forces yields interesting results: the acoustic radiation force, due to its outward-pointing nature, resists cell concentration. Meanwhile, the drag force, thanks to acoustic streaming, propels cells inward to the center. Crucially, our device operates at an ultralow frequency of approximately 109 kHz. Under such conditions, the acoustic radiation force is substantially less impactful on cell movement compared to the drag force. Therefore, we posit that in our device, it's primarily the acoustic streaming mechanism that drives cell concentration to the center.

Per the reviewer's comment, we have updated the related discussion in the manuscript.

Figure S13 | Simulation results depicting the acoustic pressure amplitude and acoustic streaming velocity on a cross-sectional plane near the device bottom.

Changes to the manuscript (Page 15 in the main text):

For the acoustic mechanism, in most of object manipulations by acoustic technologies, both the acoustic radiation force and drag force from acoustic streaming play a significant role [66-68]. However, for our AIMS platform, it is the drag force that primarily facilitates cell concentration towards the device center. We provided visual aids in the form of Fig. S13 to further clarify this. Fig. S13a illustrates the distribution of the acoustic pressure amplitude, indicating a higher value compared to surrounding areas. This creates an acoustic radiation force that radiates from the center towards the outer regions. Conversely, Fig. S13b highlights the creation of acoustic streaming by these waves, where a low velocity region in the center is evidenced. The flow lines of acoustic streaming extend from the periphery to the center, assisting in the transportation of cells to the central area, consequently forming a rounded cell cluster. The interplay of these forces yields interesting results: the acoustic radiation force, due to its outward-pointing nature, resisted cell concentration. Meanwhile, the drag force induced by acoustic streaming propelled cells inward to the center. Crucially, our device operated at an ultralow frequency of approximately 109 kHz. Under such conditions, the acoustic radiation force was substantially less impactful on cell movement compared to the drag force. Therefore, in our device, it was primarily the acoustic streaming mechanism that push cells concentrate to the center. Therefore, our AIMS platform can maintain cells in their general culture environment and simplify collection for downstream biomedical studies.

Comment 4:

Figure 4c the viability is presented on the scale of 0-12 and not as a percentage of the population please clarify this.

Response 4:

The updated macrophage viability was presented in Fig. S9 now. We used CCK-8 assay to measure the absorbance of each group. The absorbance represented the viability of macrophages. So typically, when using cells for a temporary experiment, we could describe the percentage of a control group to explain cell viabilities. However, in this experiment, macrophages were cultured for a couple of days and macrophages have a high proliferation rate. Therefore, using percentage of population is not suitable.

Figure S9 | CCK-8 assay measuring the viability of Human THP-1 cells from different groups after incubating for 1 and 2 days. Data are graphed as the mean \pm SD (n=4, biological repeats). **p < 0.01, ***p < 0.001.

Comment 5:

The simulations presented here are not described in sufficient detail in the method section. Please provide information of which boundary conditions was used, and how the excitation was done such that they can be replicated. For this the details of the geometry and dimensions are also needed.

Response 5:

Per the reviewer’s comment, we have expanded our methods section to include a more detailed account of our simulation processes. We have also added an illustration diagram in Fig. S14.

Figure S14 | Illustration of the geometry and dimensions of the simulation model.

Changes to the manuscript (Page 17 in the main text):

In brief, the model setup consisted of a Ring Piezoelectric Transducer (PZT) located at the base of a glass substrate, affixed with a thin layer of Epoxy. Positioned atop this glass substrate was a small droplet. The table below delineated the geometric parameters of our model for a better understanding:

$R1$	$R2$	$R3$	$h1$	$h2$	$h3$	$h4$	d
7	9	3.22	1.2	0.2	0.2	0.9	6.5

(Geometric parameters (all units in mm))

For the calculation of the acoustic pressure distribution, we coupled the Electrostatics, Solid Mechanics, and Thermoviscous Acoustics modules using default interfaces in Comsol Multiphysics 5.6. We assigned a ground terminal to the upper surface of the PZT, whereas the lower surface received an Electric potential of 30V. The droplet's outer surface was treated with a slip impedance of air. This part of the study was conducted in the frequency domain at 109.2 kHz. In the subsequent phase of the study, intended to compute acoustic streaming, we applied the body force⁷⁴ derived from the preceding acoustic pressure analysis to the Laminar Flow module. For

aiding module convergence, we constrained this module with a pressure point of 0 Pa on the bottom surface.

Comment 6:

The heatmap in figure 2b shows that some of parts of the secretome is down regulated when comparing the 30 min to the 60 min conditions. This links to the discussion I would like to see about what constitutes an improved secretome. Are these proteins less important for the quality/function of the secretome?

Response 6:

We appreciate the reviewer’s helpful comment. In the first version of the manuscript, the cytokine array assay was only performed one time (presented as a heatmap image). In addition, the expression period of different cytokines may be different. Per the reviewer’s comment, we prepared four repeats for each group and collected them together. Then we used the collected medium to perform the cytokine array assay. We also extended the incubation time to four days. The result is shown below:

Figure 2 | b, Heatmap presenting the secretome profiles of different groups after incubating for 4 days. The data was processed by the Z-score normalization and clustering analysis.

There were still a few abnormal targets (such as IL-3 and IL-12) that showed inconsistent tendency. There are several potential reasons: first, the heatmap results come from the single integrated densities of each targets, although we collected four repeats together for each group. Second, the accuracy of a cytokine array assay is not as good as an ELISA assay. Therefore, using ELISA assay to detect more cytokines will largely support the results. We detected 8 targets by ELISA assay and tried to describe the results more accurately. These eight targets include growth factors, chemokines, pro- and anti-inflammatory factors. We have repeated these ELISA assays after functional blocking of N-cadherin and priming MSCs by IFN- γ . For the term "improved secretome", in different situations it may have different definitions (*Engineering the MSC Secretome: A Hydrogel Focused Approach, Adv. Healthcare Mater. 2021, 10, 2001948*). Typically, most of studies prefer to use it to describe a higher concentration of the secreted proteins. However, it may be changed if the paper is describing a specific scenario, for example, lung disease, wound repair, cancer progression, etc. In our manuscript, "improved secretome" indicates the higher concentrations of overall expression of cytokines. We have modified the related description in the section of "MSC secretome was improved after acoustofluidic assembly". Indeed, in Fig. 4, we used IFN- γ to pretreat MSCs to make the secretion profile more suitable for immunomodulation therapy. Since the overall expression of cytokines was upregulated, priming by IFN- γ could help MSCs to further secrete more anti-inflammatory cytokines and control the concentration of pro-inflammatory cytokines. For immunomodulation applications, the anti-inflammatory factors such as IL-4, IL-10, and several growth factors play a more important role. Therefore, according to our results, the secretion of anti-inflammatory factors was improved, and the pro-inflammatory response of immune cells was inhibited.

Changes to the manuscript (Page 6 in the main text):

The concentrations of a broad spectrum of soluble proteins comprised of 80 total targets (78 available), such as growth factors and chemokines, were investigated using a cytokine assay to analyze the cytokine secretion profiles (Fig. 2b). For the 60 min group, 57 targets had the highest concentration compared to other groups, and 14 targets had the second-highest concentration. For the 30 min group, 14 targets had the highest concentration, and 44 targets had the second-highest concentration. However, for the 10 min group, only 6 targets had the highest concentration and 10 targets had the second-highest concentration. As for 3D Microplate group, after normalizing to the same DNA content, only 1 target had the highest concentration and 9 targets had the second-highest concentration. In addition, 65 targets had the lowest concentration in the control group. In general, the groups of 3D Microplate, 10 min, 30 min, and 60 min improved the secretome profile of MSCs, compared to the control group.

Comment 7:

I would like to say I'm not able to critically review all aspects of the in-depth immunology presented in this paper. This concerns primarily figure 4 and the selection of which proteins that are important to include in the panels and whether or not the pro-inflammatory macrophages is the most relevant choice for testing the secretome.

Response 7:

We appreciate the reviewer's helpful comment. To better illustrate the immunomodulation activity, in addition to macrophages, human T cells and B cells were added. We studied the activation of T cells, proliferation and apoptosis of B cells, and the cytokine secretion of B cells. For all these cell types, we used LPS, activation beads and antibody cocktails to mimic the pathogen activation. We used the collected conditional medium to incubate these cells and observe whether the immune of these cells can be inhibited. Theoretically, anti-inflammatory factors will play the most important role. The results are shown below:

Figure 4 | Harnessing the enhanced MSC secretome for immunomodulation applications. **a**, Heatmap measuring the inflammatory factors secreted from IFN- γ primed MSCs from different groups. **b-i**, ELISA assay measuring the secretion of FGF-2, IGF-1, VEGF, HGF-1, IL-6, IL-10, IFN- γ , and TNF- α from different groups after incubating for 4 days. **j**, Flow cytometry analysis measuring the percentage of CD86 and CD163 positive human THP-1 cells after incubating with the conditional medium from MSCs for 2 days. **k**, Flow cytometry analysis measuring the percentage of CD4 positive T cells after incubating with the conditional medium from MSCs for 3 days. **l**, Flow cytometry analysis measuring the B cell proliferation after incubating with the conditional medium from MSCs for 7 days. **m-n**, ELISA assay measuring the secretion of TNF- α and IL-10 from B cells. Data are graphed as the mean \pm SD (n=4), *p < 0.05, **p < 0.01, ***p < 0.001, ****p < 0.0001.

According to our results, we found for T cells, the percentage of CD4 positive cells of control, control + IFN- γ , 60 min and 60 min + IFN- γ groups were 92.1, 84.7, 74.5, and 72.3%, respectively. However, for CD8, the differences among all groups were not very significant. The 60 min + IFN- γ groups could still keep 91.4% of CD8 positive cells. Therefore, the MSC secretome was more effective at regulation of CD4 positive T cells than CD8 positive T cells. As for B cells, for cell proliferation, compared with activated B cells, the cells in the conditional medium from the control and control + IFN- γ groups were slightly decreased. The 60 min group was lower than the control and control + IFN- γ groups. The 60 min + IFN- γ group had the lowest number. These results indicate that the enhanced MSC secretome can effectively inhibit activated B cell proliferation. However, for cell apoptosis, all groups had no clear difference. The secretion of TNF- α also had no significant differences among all groups. However, for IL-10, the 60 min +

IFN- γ group had the highest concentration. All the control, control + IFN- γ , 60 min, and 60 min + IFN- γ groups had higher concentrations than the activated B cells without conditional medium incubation ($p < 0.0001$). In conclusion, although there is some inconsistency in our results - for example, some of the pro-inflammatory cytokines from MSCs still have comparable secretion level among all groups and the percentage of CD8 positive T cells and B cell apoptosis had no clear tendency - it is still reasonable to conclude that the MSC secretome enhanced via acoustofluidic assembly shows a more substantial immunomodulation effect than traditional MSC cultures.

Changes to the manuscript (Page 13 in the main text):

Subsequently, we also examined whether the enhanced MSC secretome can modulate the functions of T cells and B cells. For T cells, we used a commercialized Human T-Activator CD3/CD28 beads to activate T cells. As shown in Fig. 4k and Fig. S11, 97.7% of cells expressed the CD4 marker and 97.8% of cells could express the CD8 marker. Then the cells were incubated with the conditional medium of each group. The percentage of CD4 positive cells of control, control + IFN- γ , 60 min and 60 min + IFN- γ groups were 92.1, 84.7, 74.5, and 72.3%, respectively. The tendency was similar to CD86. However, for CD8, the differences among all groups were not very significant. The 60 min + IFN- γ groups could still keep 91.4% of CD8 positive cells. Therefore, the MSC secretome was more effective at the regulation of CD4 positive T cells compared to CD8 positive T cells. As for B cells, we used an antibody cocktail (F(ab)2 anti-IgM, IL-2, and anti-CD40 antibody) to stimulate original human B cells according to previous studies 56,57. Then the cells were incubated with the conditional medium of each group for 7 days. Then the cell proliferation and apoptosis were studied. As shown in Fig. 4l and Fig. S12, for cell proliferation, compared with activated B cells, the cells in the conditional medium from the control and control + IFN- γ groups were slightly decreased. And the 60 min group was lower than the control and control + IFN- γ groups. The 60 min + IFN- γ group had the lowest number. These results indicated that the enhanced MSC secretome could effectively inhibit activated B cell proliferation. However, for cell apoptosis, all groups had no clear difference. At last, we used ELISA assay to measure the cytokine secretion capacity of B cells from different groups. As shown in Fig. 4 m and n, the secretion of TNF- α had no significant differences among all groups. However, for IL-10, the 60 min + IFN- γ group had the highest concentration. All the control, control + IFN- γ , 60 min, and 60 min + IFN- γ groups had higher concentrations than the activated B cells without conditional medium incubation ($p < 0.0001$). In conclusion, although there was some inconsistency in our results - for example, some of the pro-inflammatory cytokines from MSCs still have comparable secretion level among all groups and the percentage of CD8 positive T cells and B cell apoptosis had no clear tendency - it is still reasonable to conclude that the MSC secretome enhanced via acoustofluidic assembly shows a more substantial immunomodulation effect than traditional MSC cultures.”

REVIEWERS' COMMENTS

Reviewer #1 (Remarks to the Author):

In this revised version of the manuscript the Authors addressed the raised comments satisfactorily. Therefore it can be published in the present format

Reviewer #2 (Remarks to the Author):

I appreciate the efforts you have made to improve your work. Your study now shows a clear structure, a sound methodology and a convincing argumentation. You have taken into account most of my comments and recommendations and discussed your findings adequately.

Reviewer #3 (Remarks to the Author):

Dear Authors,

I'm happy with the responses to the issues raised in my previous review and I find that the manuscript is significantly improved.

I would recommend that it is accepted for publication.